# A matched case-control analysis of autonomous vs human-driven vehicle accidents

Mohamed Abdel-Aty[1] & Shengxuan Ding [1] ✉

Despite the recent advancements that Autonomous Vehicles have shown in their potential to improve safety and operation, considering differences between Autonomous Vehicles and Human-Driven Vehicles in accidents remain unidentified due to the scarcity of real-world Autonomous Vehicles accident data. We investigated the difference in accident occurrence between Autonomous Vehicles' levels and Human-Driven Vehicles by utilizing 2100 Advanced Driving Systems and Advanced Driver Assistance Systems and 35,113 Human-Driven Vehicles accident data. A matched case-control design was conducted to investigate the differential characteristics involving Autonomous' versus Human-Driven Vehicles' accidents. The analysis suggests that accidents of vehicles equipped with Advanced Driving Systems generally have a lower chance of occurring than Human-Driven Vehicles in most of the similar accident scenarios. However, accidents involving Advanced Driving Systems occur more frequently than Human-Driven Vehicle accidents under dawn/dusk or turning conditions, which is 5.25 and 1.98 times higher, respectively. Our research reveals the accident risk disparities between Autonomous Vehicles and Human-Driven Vehicles, informing future development in Autonomous technology and safety enhancements.

Automation of systems has been experiencing rapid development and has brought about a revolution in the transportation industry. The introduction of Autonomous Vehicles (AV) technology has made the vision of a safe transportation system with effortless driving seem attainable. It is anticipated that the automation of systems will significantly reduce the number of accidents, as human errors contribute up to 90% of accidents[1]. While smart transportation has showcased several benefits, these emerging technologies have also exhibited drawbacks, particularly regarding safety risks. Accidents of on-road testing have already been documented in limited testing data[2].

According to findings from the RAND corporation[3], a mere advancement in safety features in the initial release of AVs could yield significant life-saving results. The research suggests that if AVs were to be introduced with an average safety level ten percent higher than that of the typical human driver, approximately 600,000 fatalities could be averted in the United States over a span of 35 years. Nevertheless, it is crucial to acknowledge that between 2015 and 2022, there was a yearly rise in both the annual miles traveled by autonomous vehicles (AVMT) and the number of AV accidents on public roads in California, with the exception of a decline in 2020 attributable probably to the COVID-19 pandemic[4]. AV testing on California public roads is permitted by the California Department of Motor Vehicles (CADMV)[5]. Up until June 2023, we have identified and enriched with 598 Advanced Driving Systems (ADS (SAE Level 4)) accidents in California (AVOID dataset)[6]. These reports provide details about AV accidents and disengagements, which happen when the vehicle switches from autonomous mode due to technological issues or when the test driver or operator assumes manual control for safety purposes[7]. The detailed information included in these reports revealed various factors related to AV accidents.

[1]Smart and Safe Transportation Lab (SST), Department of Civil, Environmental and Construction Engineering, University of Central Florida, 12800 Pegasus Dr, Orlando, FL 32816, USA. ✉e-mail: shengxuan.ding@ucf.edu

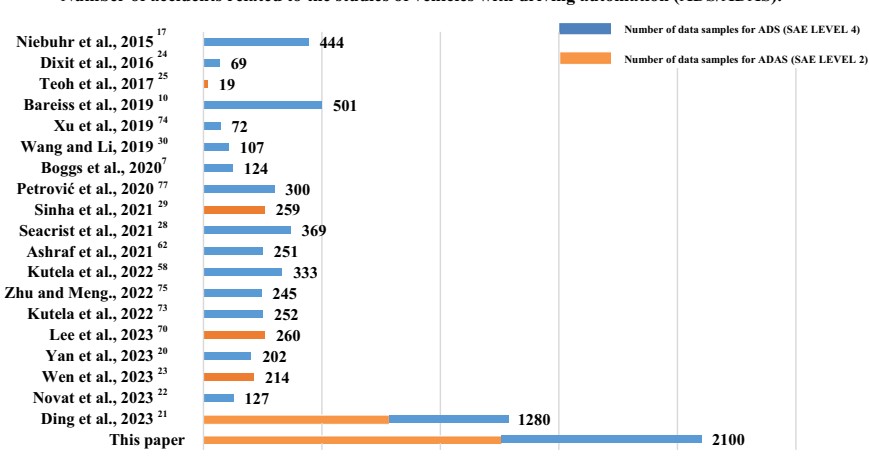

**Fig. 1 | Number of accidents related to the studies of vehicles with driving automation (ADS/ADAS).** The blue line shows the number of ADS data samples, and the orange line shows the number of ADAS data samples used in related studies.

We further collected the corresponding reports of Human-Driven Vehicles (HDV) accidents to contrast the differences between AV and matched HDV accidents.

There are many potential benefits of AVs on traffic safety, such as a reduction in human error, reduced fatigue, and distraction. ADAS functions including Electronic Stability Control (ESC), Anti-Lock Braking (ABS), and Information and Communication Technology aids in ongoing driving tasks to prevent accidents[8–10]. For example, Tesla Autopilot consistently maintains a closer distance to the lane center than human drivers[11]. Additionally, Level 3 or higher levels of automation can further enhance traffic by reducing accidents[12,13], improving mobility for the disabled and elderly, and minimizing traffic collisions through efficient driving and the reduction of human errors[14]. Koopman and Wagner highlighted the importance of understanding how AVs will interact with human drivers, pedestrians, and other road users[15]. Compared with HDV, accident avoidance features of AVs can reduce accidents and fatalities caused by distracted driving or human error by helping control the vehicle and alerting drivers to potential dangers[16,17].

However, there are also some possible safety challenges that need to be addressed in AVs. Penmetsa et al.[18] identified several safety challenges that need to be addressed in the development of AV, such as the need for reliable sensing and perception, robust decision-making algorithms, and fail-safe mechanisms[18]. Kalra and Paddock presented a statistical methodology with 382 accidents per 100 million miles based on binomial and Poisson distributions for estimating the required mileage to establish the reliability of AVs, it shows that AVs would have to be driven hundreds of millions of miles and sometimes hundreds of billions of miles to demonstrate their reliability in terms of fatalities and injuries[19]. Some studies also have concentrated on analyzing the accident severity involving different types of AV[20]. Ding et al.[6] collected 1280 cases to compare factors related to injury severity between the ADAS and ADS accidents by random parameter multinomial logit models. Subject vehicle's contact area, road and environment, and pre-accident movement, significantly impact accidents injury severity. ADS-equipped vehicles in work zones have a higher probability of being involved in minor and moderate/severe injury accidents[21].

The comparison of safety performance between AV and HDV is a topic of debate, with conflicting viewpoints[22]. On one side, numerous studies support the view that AVs are generally safer than HDVs[23]. For instance, Dixit et al.[24] analyzed the statistical distribution of reports on 69 manual disengagements and accidents, they compared the accident rates of Google self-driving cars to those of human drivers and found that fatal accidents involving Google cars were lower than that of HDV, no fatalities occurred as compared to 1 death for every 108 million miles in California[24]. Additionally, from 2009 to 2015 in Mountain View, California, Google cars demonstrated a significantly lower rate of police-reportable accidents per million vehicle miles traveled (VMT) compared to human drivers (2.19 versus 6.06)[25]. On the other hand, some research challenges this view, suggesting that the safety of AVs may not always exceed that of HDVs. Schoettle and Sivak uncovered that AVs have a higher rate of accidents per million miles traveled compared to HDVs in limited (and generally less demanding) conditions (e.g., avoiding snowy areas)[26]. Favarò et al.[27] found that rear-end accidents were the most frequent type of AV accidents, with AVs being hit from behind by conventional vehicles at a rate twice that of rear-end accidents than rear-end "fender-benders" for conventional vehicles in California 2013[27].

These studies offered significant insight into the factors that contribute to AV and HDV accidents, while earlier studies may not consider sufficient factors of road environment, accident outcome, pre-accident condition, and accident type for accident analysis due to a lack of matched AV and HDV data for identifying the characteristics of AV accidents and how they differ from regular HDV accidents[28–30]. The number of accidents of the relevant studies and their automation level are shown in Fig. 1.

In this work, we examined a dataset comprising both AV and HDV accidents. The dataset utilized in this study was compiled from various sources and encompassed information on accident types, road and environmental conditions, pre-accident vehicle movements, and accident outcomes. We use a matched case-control logistic regression model. We further add the National Highway Traffic Safety Administration (NHTSA) AV database comprising an additional 495 ADS and 1001 Advanced driver assistance systems (ADAS (SAE Level 2)) accidents[31]. These findings illuminate the factors that contribute to accidents involving AVs vs. HDVs.

## Results

### General trends in the full accident data

We first present general trends and comparisons between AV and HDV accidents of the full dataset. Figure 2a–d displays the distribution of factors affecting AV (2100), including SAE level 4 ADS (1099), level 2 ADAS (1001) and HDV accidents (35,133), respectively. Notably, vehicles make up 80% of participants in AV accidents, with pedestrians accounting for 3%. In contrast, for HDVs, pedestrians constitute 15%

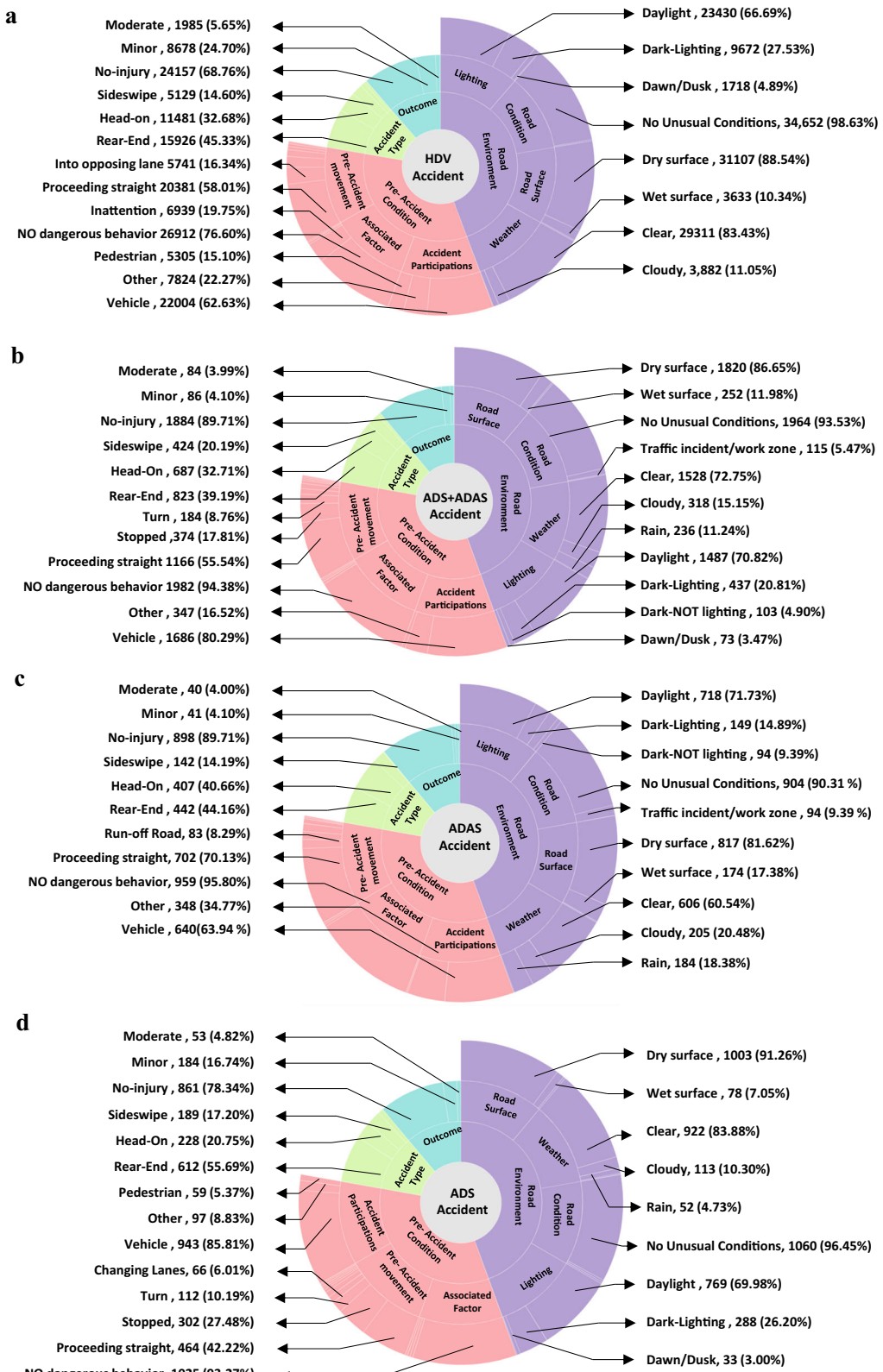

**Fig. 2 | Distribution of the factors influencing accidents of various vehicle types. a** HDV accidents with a sample of 35,133. **b** SAE level 4 ADS + SAE level 2 ADAS accidents with a sample of 2100. **c** SAE level 2 ADAS accidents with a sample of 1001. **d** SAE level 4 ADS accidents with a sample of 1099.

and vehicles 63% of accident participants, as depicted in Fig. 2a, b. When examining the outcomes of accidents, both AVs and HDVs lead to either no injuries or minor injuries occurring in 94%.

Significant disparities between AV and HDV accidents can be seen in work zones, traffic events, and pre-accident movements such as

slowing down, proceeding straight, and moving into opposing lanes, with AVs exhibiting higher accident rates. For both AVs and HDVs, the most frequent pre-accident movement is proceeding straight. It is observed that 56% of AV accidents and 58% of HDV accidents occur under this specific condition. About 5% of AV accidents take place in

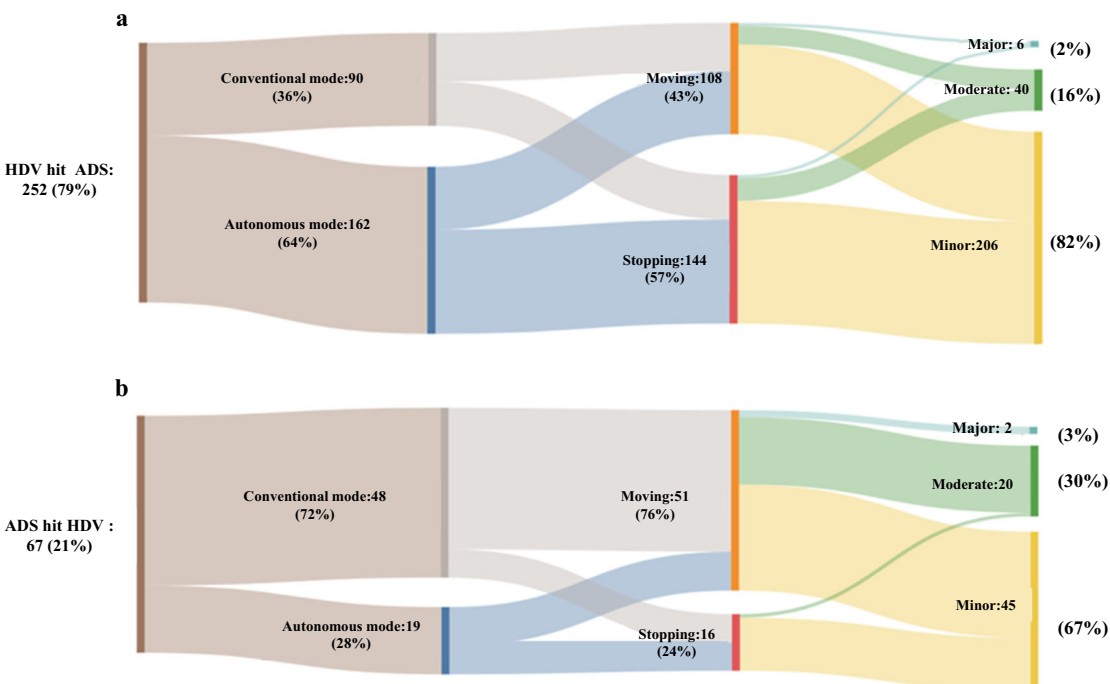

**Fig. 3 | Rear-end accident conditions between ADS and HDV. a** Rear-end accidents that HDV hit an ADS from behind with a sample of 252. **b** Rear-end accidents that ADS hit an HDV from behind with a sample of 67.

locations impacted by previous traffic events or work zones, where normal traffic is disrupted by earlier incidents like disabled vehicles or spilled cargo. In comparison, just 1.3% of HDV accidents occur in similar settings. Analyzing the pre-accident scenario, a distinctive observation is that only 1.8% of AV accidents are attributed to inattention or poor driving behavior, in contrast to a much higher 19.8% for HDVs.

Evaluating environmental factors, the majority of accidents involving both AVs and HDVs tend to happen under clear weather conditions. Notably, accidents involving HDVs occur slightly more frequently under these conditions, at a rate of 83%, compared to 73% for AVs. However, AVs are more commonly involved in accidents during rainy conditions, accounting for 11% of such accidents, compared to HDVs which experience these conditions in only 5% of accidents. Dawn or dusk conditions experience 3.5% of AV accidents, which is lower than the 4.9% rate for HDVs.

In terms of accident type, rear-end accidents constitute a majority for both AVs and HDVs. Furthermore, we determine the accident type associated with AV and HDV based on the accident angle. Our data indicates that for HDVs, the rear-end stands at 45% (other vehicles hit HDV), while head-on accidents (HDV hit other vehicles) occur at a rate of 33%. In contrast, AVs have a slightly lower rear-end accidents (other vehicles hit AV) rate of 39%, but a similar head-on accidents (AV hit other vehicles) rate of 33%. This suggests that while AVs have a marginally lower percentage of being rear-ended compared to HDVs, their percentage of head-on accidents is almost the same.

Figure 3 describes two conditions related to ADS rear-end accidents: a: HDV has hit an ADS from behind (252) and b: an ADS has hit an HDV from behind (67). The left side of the diagram starts with two conditions of HDV hit ADS or ADS hit HDV. The middle section shows the movement of vehicle: Moving or Stopping. The right side categorizes the severity of the accidents: Minor, Moderate and Major. The numbers indicated in each section of the diagram correspond to the total count of each specific category. The width of each link connecting the sections of the diagrams represents the proportion of scenarios that fall into the subsequent category. For example, In Fig. 3a, which details accidents where a HDV hit an ADS, we see that for the accidents where the ADS was moving (middle section), 62 cases (57%) involved ADS in autonomous mode while 46 cases (43%) involved ADS in conventional mode (left section). Among the 108 accidents categorized under the 'moving condition', there were 2 cases (2%) that resulted in major injuries, 18 cases (17%) that led to moderate injuries, and the remaining 88 cases (81%) involved minor injuries.

The analysis reveals that 79% of rear-end accidents involve HDV hitting ADS, while 21% of rear-end accidents involve ADS hitting HDV. When the ADS hit HDV in conventional mode, most of the ADS are moving. We may conclude that compared with the autonomous mode, human drivers may not react as quickly or may not notice the object in time to take appropriate action. In terms of accident severity, 206 of accidents (82%) occur as minor injury when HDV hit ADS. This percentage is 67% when the ADS hit HDV. It is important to note that a majority of moderate and major accidents involving an ADS hitting an HDV occur when both vehicles are moving in the conventional mode. Notably, in cases where HDVs hit ADS, 64% of ADS are operating in autonomous mode. Conversely, when ADS are responsible for hitting HDVs, 72% of ADS are operating in the conventional mode. According to Dixit et al. in 2016, 56.1% of disengagements were attributed to system failures, 26.57% were initiated by the driver, and 9.89% were related to road infrastructure issues[24]. This observation suggests that conventional mode occurred more frequently than autonomous mode where ADS hit the HDV. This may be attributed to the advance autonomous mode of ADS. Autonomous mode uses advanced algorithms to detect and avoid obstacles and other vehicles in the path of the vehicle[32].

We also contrast the ADS vs the ADAS, accidents related to ADAS and ADS display differences across various conditions, as shown in Fig. 2c, d. Regarding weather and road conditions, ADAS has 23.34% fewer accident number in clear skies but a 13.65% higher in rain compared to ADS. For road conditions, ADAS accidents experience a 7.48% higher accident number in traffic events or work zones and 10.33% higher accident number on wet roads than ADS. Analyzing pre-accident movements, ADAS indicate a 27.91% higher accident number for proceeding straight, while reporting 3.0% fewer turning accident numbers than ADS. In terms of accident types, ADAS is 3.0% higher accident number than ADS in broadside accidents and lags by 5.4% in sideswipe accidents. From an injury outcome perspective, ADAS

**a**

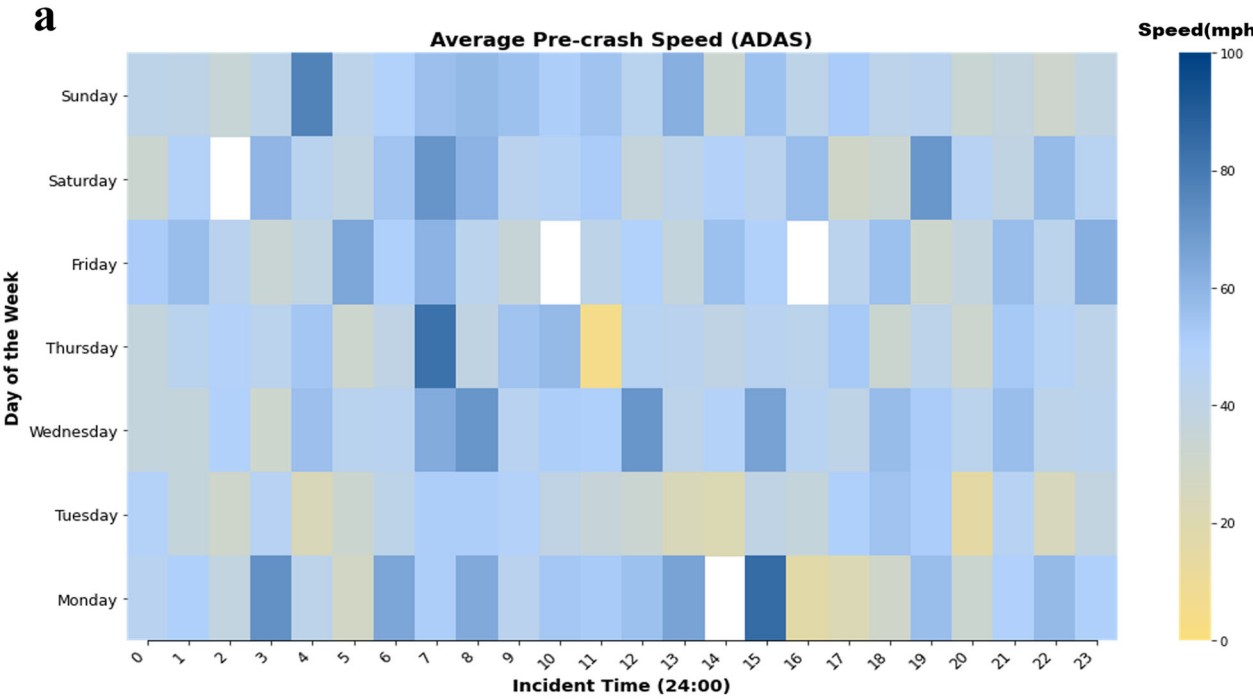

**b**

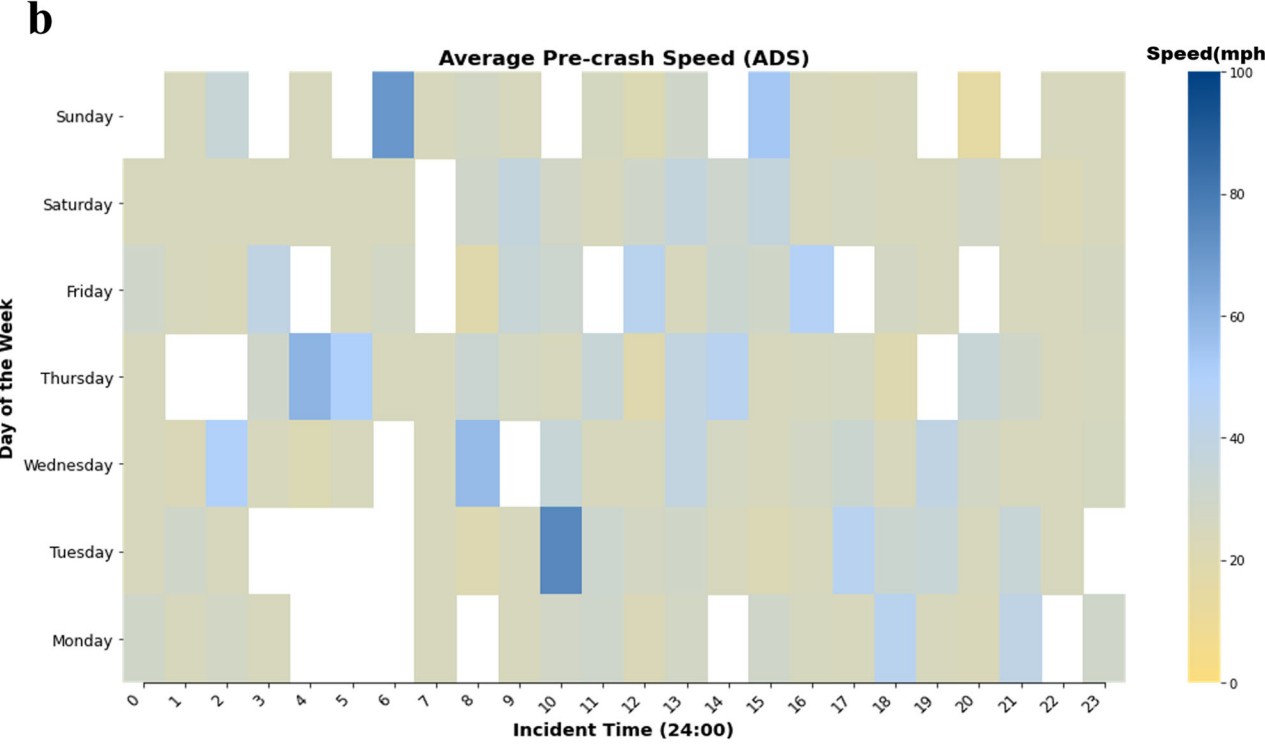

**Fig. 4 | Distribution of the pre-accidents Speed. a** ADAS Average Pre-accident Speed Heatmap. **b** ADS Average Pre-accident Speed Heatmap.

accidents present a 11.37% higher accident number of no-injury but a 2.1% lower number in fatal injuries against ADS. To enhance the understanding of pre-accident speeds, heatmaps that visually represent these speed patterns for ADS and ADAS vehicles are shown in Fig. 4. This chart offers a detailed comparison of how speeds vary across different days of the week and at various times of the day. This trend can be attributed to the fact that ADAS is primarily designed for

highway use, leading to a higher average pre-accident speed when compared to ADS vehicles, which are designed for a complex urban driving scenario.

We have also analyzed the full data to identify the influence of roadway elements and factors related to time using a random parameter logit model[33]. Only a single random parameter "Day of the week" demonstrated a significant effect. Upon analyzing the model, we found

that the dawn/dusk and turn conditions exhibit positive coefficients that are statistically significant at a 95% confidence level. This indicates a higher odd of an AV accident occurrence when these variables are significant in the random parameter logit model. Furthermore, we discovered that several variables demonstrate high significance and exhibit negative coefficients, suggesting a reduced likelihood of an accident when these factors are significant. These variables include the rain conditions, rear-end conditions, broadside conditions (a broadside condition is a car accident that occurs when the front of one vehicle slams into the side of another vehicle), moderate severity, proceeding straight, run-off road, backing, and entering traffic lane.

### Findings of road, environment, and accident type

Based on the results of the matched case control logistic regression model, compared with HDV, the odds of an ADS accident occurring in rainy weather are 0.335 times. This indicates a lower likelihood of an ADS accident in rainy weather compared to an HDV accident. RADAR is capable of detecting objects at distances exceeding 150 m, even in adverse weather conditions such as fog or rain[34]. In contrast, human drivers may only be able to perceive objects up to approximately 10 meters away under similar circumstances[35]. Although adverse weather can increase the likelihood of potential failures or loss of sensors[36–38], recent innovations in visual algorithms, coupled with the combined use of cameras, LIDAR, GNSS, and RADAR sensors[39,40], are crafted to recognize pedestrians and vehicles under varying weather scenarios, such as cloudiness, snow, rain, and darkness[41,42]. This offers solutions to the challenges associated with driving in less-than-ideal conditions. In contrast, human drivers may have difficulties seeing through heavy rain or fog, leading to a delay in detecting potential hazards or reacting appropriately[40].

Interestingly, the dawn/dusk odds ratio indicates a 5.250 higher probability of ADS accident than HDV accident. This could be attributed to the sensors and cameras used by AVs may not be able to quickly adapt to changes in lighting conditions, which could affect their ability to detect obstacles, pedestrians, and other vehicles[39,43]. At dawn and dusk, for instance, the sun's shadows and reflections may confuse sensors, making it hard for them to distinguish between objects and identify potential hazards. Furthermore, the fluctuating light conditions can impact the accuracy of object detection and recognition algorithms used by AVs, which can result in false positives or negatives[35,44].

Accident types related findings for ADS and HDVs are worth noting. Compared to HDV accidents, AVs experience relatively lower risks in rear-end and broadside accidents (0.457 times and 0.171 times, respectively). This finding indicates that AVs may detect and react to potential rear-end and sideswipe accident situations much faster than humans can. This is because they are equipped with advanced sensors and software that can quickly analyze the surrounding environment and make decisions based on the data received[45]. In addition, the kinematic method used by ACC system keeps track of and regulates the distance between vehicles, alerting drivers if this space becomes smaller than the safe limit, especially at highways[46]. By ensuring that vehicles keep a consistent speed and distance between vehicles, thus effectively reduces the risk of rear-end accidents[47]. Compared with ADS, HDVs tend to display greater velocity differences at larger spacing ranges[48], a factor that significantly contributes to a higher incidence of rear-end and sideswipe accidents[49].

### Findings of pre-accident conditions and accident outcomes

In terms of pre-accident conditions, most of the pre-accident movements made by ADS reduce the probability of accidents from the results of the matched case control logistic regression, except for turning, which increases the likelihood of an accident by 1.988 times compared to HDVs. One possible reason is a lack of situational awareness. Situational awareness of AVs can be defined as the ability of these vehicles to perceive essential elements in their surroundings,

understand the importance of these elements, and anticipate their future state or changes[50]. The complexity of turning in autonomous driving scenarios arises from three primary challenges: choosing the appropriate lane (target lane selection), devising and computing a safe and efficient path (trajectory planning and calculation), and executing the turn while adjusting to dynamic conditions (vehicle controlling and tracking)[51]. AVs rely on sensors and algorithms to perceive their surroundings and make driving decisions[45]. However, these systems may not detect all obstacles and hazards, particularly in complex and dynamic driving scenarios like turning at intersections[52,53]. It is a significant challenge to generate sufficient information and achieve comprehensive detection of the surrounding environment from a single independent source due to limited sensor ranges and limited coverage of the environment by sensors in AV[45,54]. Additionally, some AVs are programmed to follow predefined rules and scenarios, which may not encompass every possible driving situation[55–57]. The modifications of scenarios can present difficulty for AVs in perceiving and responding to them, thereby raising the risk of an accident[58]. Moreover, multiple oncoming HDVs and the complexity of such driving scenarios are a considerable challenge for AVs such as unprotected left turns at intersections[59]. These situations are complicated by factors like limited priority and variation in trajectories[60]. AVs tend to be overcautious (such as having a longer startup delay during the turning at intersections)[61], which can lead to rear-end or sideswipe accidents with HDVs[62]. Furthermore, multi-interactions caused by mixed flows aggravate uncertainties in detection, such as the superposition of distance and angle measurement error[63]. Conversely, HDVs can adapt and modify their speed more seamlessly than AVs, highlighting the limitations of AVs in comparison to the adeptness of experienced drivers[64]. And AVs face difficulties with executing lane changes or turning in heavy traffic and lack psychological insight[65,66]. In addition, HDVs can predict pedestrian movements and exercise caution based on their driving experience, whereas AVs may struggle with recognizing pedestrians' intentions, potentially leading to emergency braking or accidents due to a lack of understanding of social cues and psychological reasoning[65,67,68].

ADS accidents are less likely to occur than HDV accidents in situations such as proceeding straight, run-off road (a vehicle leaves the designated roadway and travels onto an area that is not intended for regular traffic) and entering traffic lane conditions (a vehicle transitioning from a stationary or parked position to enter a traffic lane and become physically present within the flow of traffic). When considering the proceeding straight condition, it was found that AV accident resulted in a 0.299 lower probability of an HDV accident. Remarkably, ADS accident risk is 0.021 times as high as that of an HDV accident in run-off road condition, which can be explained by the faster reaction time of AVs[24]. AVs can detect these situations and apply corrective actions, such as adjusting the speed or steering angle[69,70], more quickly and accurately than a human driver[71,72]. The result of matched-case control model revealed a significant correlation between the entering traffic lane condition of ADS accident, the risk of which is 0.267 times as high as HDV accident. According to the results of the matched case-control logistic regression, the impact of backing is noteworthy, which shows that the ADS is less likely to be affected than the HDV. According to the analysis, the model using accidents of AVs showed a decreased probability of accidents for moderate and fatal severity in comparison to HDV.

## Discussion

A comprehensive examination was performed using a dataset of AV and HDV accidents in this study. A total of 2100 AV accidents and 35,133 HDV accident records were collected, which accurately reflected the accident details. The analysis first dealt the whole available data that includes both ADS and ADAS (SAE levels 4 and 2, respectively) using general descriptive statistics, percentages, and a random

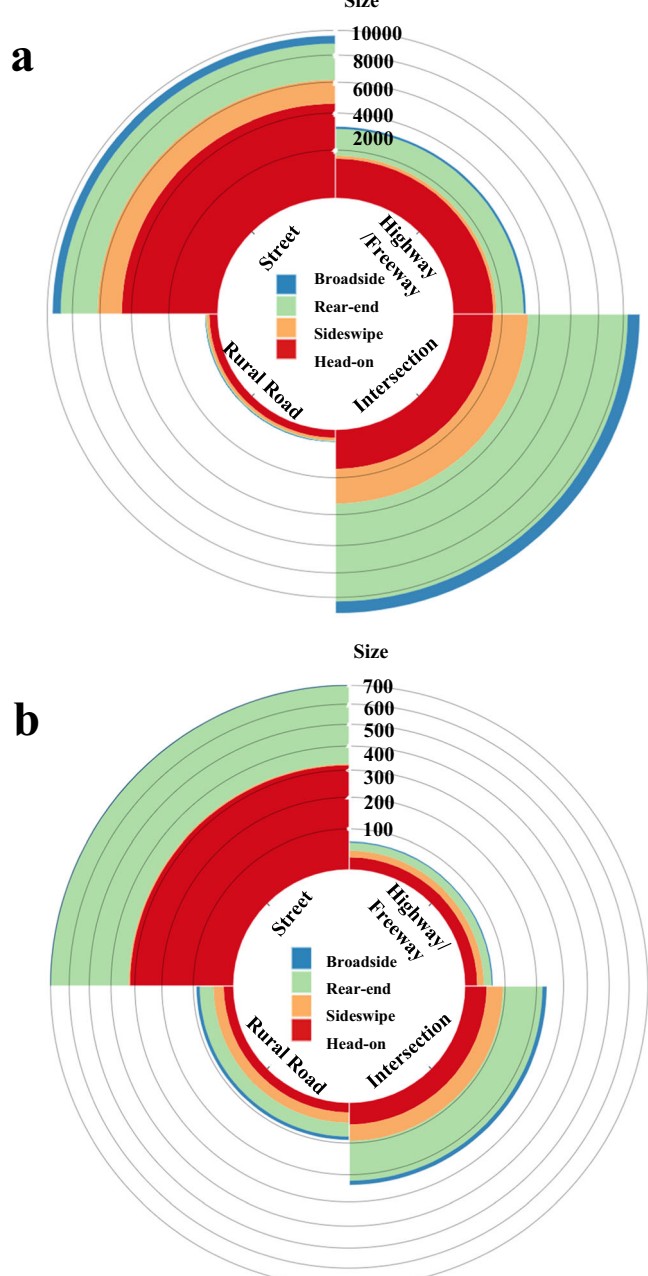

**Fig. 5 | Distribution of the accident's types and scenarios. a** HDV accidents with a sample of 35,113. **b** ADS accidents with a sample of 1099. The distributions of accident types (head-on, sideswipe, rear-end, broadside) for vehicle categories (HDV and ADS) illustrate the frequency and proportion of each accident type in the respective locations by vehicle categories.

However, the odds ratio of an ADS accident happening under dawn/dusk or turning conditions is 5.250 and 1.988 times higher, respectively, than the probability of an HDV accident occurring under the same conditions. The possible reasons might be a lack of situational awareness in complex driving scenarios and limited driving experience of AVs[21]. Improving the safety of ADS under dawn/dusk or turning conditions necessitates a holistic approach that involves advanced sensors, robust algorithms, and smart design considerations. Key strategies include enhancing weather and lighting sensors, implementing redundancy measures, and integrating sensor data effectively. By focusing on these aspects, the safety of ADS can be significantly enhanced in challenging scenarios.

Compared with current studies that only focus on AV accidents[73–75] or analyze AV and HDV with limited samples[4,76,77], this paper has analyzed the factors that contribute to AV in comparison to HDV accidents through the analysis of real-world accidents and multi-source data. Furthermore, this research encompassed both AV and HDV accidents, instead of solely concentrating on different levels of AV accidents without considering a comparison with HDV accidents. Moreover, this study addresses the issue of unbalanced data between AV and HDV accidents by employing a matched case-control study design. One of the constraints of this study is analyzing the detailed levels of AV and the specific activated ADAS or AV system in an accident. Understanding and modeling different classifications of AV versus HDV accidents can be challenging and may require more data. It would also be crucial in the future to incorporate data about right-of-way at intersections, encompassing yield signs, stop signs, priority signals, and traffic lights, to enhance the comparative analysis between AV and HDV. Future research could also benefit from consulting a group of AV experts to identify and report on the factors contributing to safety differences between HDVs and AVs. Reporting their responses could provide qualitative depth to the research findings.

## Methods
### Data preparation
The full AV data set includes 2100 (ADS + ADAS) accidents based on AVOID[6] (CADMV and NHTSA's AV accident databases[31]). Supplementary Table 3 presents the general descriptive statistics of the databases, and the description and explanation for the variables are given in Supplementary Table 4. It provides a summary of the characteristics of the available variables that are classified into four major categories in the final data. The categories of variables include road and environmental characteristics (such as weather, road condition, road surface, and lighting conditions), pre-accident conditions (including vehicle manufacturers, AV driving modes, and pre-accident vehicle movement status), accident type, and accident outcome (accident severity), as shown in Supplementary Table 3. Among this information, the day of week, time of day and road location are typical confounders relevant to the traffic accident risk[78]. To be specific, the risk of traffic accidents can vary by the day of the week and time of day due to differences in traffic volume and driver behavior (e.g., commuters mostly in peak periods). In addition, the road location can affect the risk of traffic accidents by influencing traffic volume, speed limits, and the presence of other risk factors such as road design factors, intersections, pedestrians, and bicyclists.

The second group of accidents data comprised information of HDV, which was gathered from the Statewide Integrated Traffic Records System (SWITRS)[79]. This format and structure of data can be matched with the AV data, we collected 35,113 cases of HDV accidents according to the year of AV accidents as the first step. The distributions of accident types (head-on, sideswipe, rear-end, broadside) for vehicle categories (HDV and ADS) are shown in Fig. 5, which visually illustrates the frequency and proportion of each accident type in the respective locations by vehicle categories. For HDV accidents, intersections are the primary locations (significantly higher than other HDV accident

parameter logit model (not shown in the paper for brevity and since results are almost consistent with the other matched model). The analysis considered four categories of variables, including accident type, road and environment, pre-accident movement, and accident outcomes.

The accident data of AV and HDV were compared using the matched case-control logistic regression. The impact of different variables on the potential of accidents in AV vs HDV was conducted using a matched case-control logistic regression model. Based on the model estimation results, it can be concluded that ADS in general are safer than HDVs in most accident scenarios for their object detection and avoidance, precision control, and better decision-making.

location types with $F = 5.1043$ and $p = 0.0166$), where 61.5% of HDV intersection accidents are rear-end, making it the most common type. Urban streets are the second most common scenario, with head-on accounting for 48.0% of HDV street accidents. Conversely, ADS accidents occur more frequently on urban streets (significantly higher than other ADS accident location types with $F = 10.4982$ and $p = 0.0011$), where 45.6% of ADS street accidents were head-on. Accidents at intersections are the second most common for ADS, with rear-end making up 53.8% of these ADS intersection accidents.

## A matched case-control logistic regression model

A matched case-control study is an observational study that involves comparing individuals who have a specific health outcome or disease (the cases) with individuals who do not have the health outcome or disease (the controls)[80]. The study design involves selecting cases and controls based on their exposure to a particular risk factor or characteristic, and then comparing the frequency of that exposure between the two groups. In the context of this paper, a matched case-control study could be used to investigate the relationship between accident-related risk factors[81,82]. Cases would be AVs involved in accidents, and controls would be HDVs involved in accidents.

A matched control study has been designed to investigate the impact of various factors on the likelihood of accidents in two specific scenarios: AV and HDV.

Conditional logistic regression is a variant of logistic regression that specifically tackles the issue of stratification within matched case-control studies[83]. In this research, there are $N$ strata denoted by $i = 1, 2, \ldots, N$. Each stratum has one AV accident case sample and $k$ HDV accident control samples denoted by $j = 1, 2, \ldots, k$. The conditional likelihood for the $i$ th strata depends on the probability of the total number of cases (AV accident case samples) and controls ($k$ HDV accident control samples) recorded in the stratum[84]. $P_{ri}(x_{ji})$ refers to the probability of the $j$ th samples in the $i$ th stratum is a AV accident where $x_{ij} = (x_{1ij}, x_{2ij}, \ldots, x_{pij})$ is determined by a vector of variables $(x_1, x_2, \ldots, x_p)$. A logistic regression model with linear parameters is employed to estimate the likelihood of an accident, as described by Abdel-Aty et al.[85]:

$$logit | P_{ri}(x_{ji}) = (a_i, b_1 x_{1ji}, \ldots b_k x_{pji}) \quad (1)$$

The controlled variables used to create strata are reflected in the intercept term. To incorporate the impact of stratification in the analysis, it is possible to construct a conditional log-likelihood. This log-likelihood function comprises multiple terms, each representing the conditional probability of an accident occurring within a specific stratum[86]. The following equation presents the formula for the conditional likelihood function, as stated by Abdel-Aty et al.[85]:

$$\pounds(\beta) = \Pi_{i=1}^{N} \left[ \left( 1 + \sum_{j=1}^{k} \exp \left\{ \sum_{u=1}^{p} \beta_u \left( x_{uji} - x_{u0i} \right) \right\} \right)^{-1} \right] \quad (2)$$

The coefficients' estimates in Eq. (1) are identical to the maximum likelihood function values in Eq. (2). These estimates are log-odds ratios that can provide an approximation of the relative risk of an accident and are also referred to as hazard ratios (i.e., the ratio of odds for accident occurrence versus non-occurrence). The hazard ratio is calculated by raising the exponential value to the coefficient's power. For a dummy variable, the odds ratio is a statistic defined as the ratio of the odds of the case. The odds ratio can be written as

$$OR(x_k) = \frac{\Pr(y_{i0} = 1, x_k = 1, Z) / [1 - \Pr(y_{i0} = 1, x_k = 1, Z)]}{\Pr(y_{i0} = 1, x_k = 0, Z) / [1 - \Pr(y_{i0} = 1, x_k = 0, Z)]} = \exp(\beta_k) \quad (3)$$

where, $Z$ represents the vector of explanatory variables excluding $x_k$. $\beta_k$ is estimated coefficient for $x_k$.

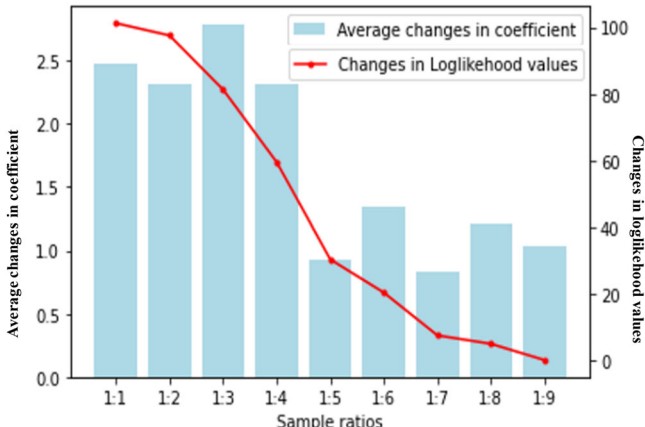

**Fig. 6 | Selection for the optimal number of controls.** The blue column line indicates the average coefficient changes (left y-axis), and the red line shows the changes in loglikelihood values (right y-axis).

## Matched case control study for ADS accidents

Our aim is to explore the differential characteristics of accidents involving AVs and HDVs, rather than comparing accidents and non-accidents. Direct comparisons between AV and HDV accidents are still not viable as the difference in exposure and number of vehicles of both types is extremely unbalanced. We incorporated Annual Average Daily Traffic (AADT) data of various road types from the California Traffic Census Program[87], which is provided in the supplementary methods. HDVs show a higher incidence of accidents on highways, intersections, and streets, particularly on highways. For rural roads, HDVs and AVs exhibit almost similar accident rates. Across all road types, HDVs consistently record significantly higher accident figures than AVs. To examine the impact of exogenous variables on accident risk for different vehicle types, we conducted a matched case-control logistic regression model for AV (ADS) and HDV accidents. The coordinates of accidents are extracted by Google Map API, and then the type of road is identified to conduct matched case-control logistic regression. The distribution of AV accidents over various situations differs from the distribution of HDV accidents is concluded from the matched case-control study.

To overcome this challenge of variables that confound the relationship between risk factors and traffic accident outcomes, the first principle is to match cases and controls at the same location. In the case of a location that does not have enough controls, similar locations within a radius of 5 miles for intersections and urban segments were used, and the day of the week and time of day were controlled. We assumed that cases and controls were under similar traffic patterns based on the controlled time and space. Aside from intersections and streets, the location of each stratum for AV and HDV accidents is on the same highways and expressways. In addition, the same road type for each stratum is controlled to ensure similar geometric design. As the manual override and conventional modes of ADAS closely resemble HDV, we only focus on ADS cases from CA for the matched case-control study. Furthermore, some cases were removed due to difficulties in obtaining or imputing precise road types for matched case-control logistic regression.

We organized the data into $N$ strata according to the occurrence of AV accidents. Each stratum consisted of one case and $k$ corresponding controls. To ensure consistency across strata, we employed a matched case-control logistic regression by adjusting the number of control samples and assessed the resulting estimates for each model. Samples generally refer to the groups of accidents selected for comparison within each stratum. Case samples are specific accidents who have the outcome that is the focus of the

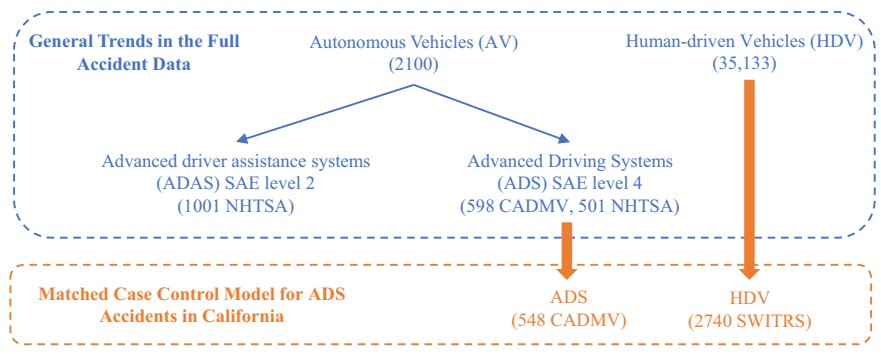

**Fig. 7 | Sample size of data.** The blue fonts indicate general accident trends, while the orange fonts represent data for the matched case control model of ADS accidents in California. The HDV data is sourced from SWITRS[79]. The ADS (SAE Level 4) data is sourced from CADMV[5] and NHTSA[31], while the ADAS (SAE Level 2) data is sourced from NHTSA[31].

### Table 1 | Matched case-control logistic regression model

| Variable | Estimated parameter | Odds Ratio | t-statistic | P value | 95% Confidence Interval of OR |
|---|---|---|---|---|---|
| *Road and environment* | | | | | |
| Rain condition (1 if weather is rain, 0 otherwise) | −1.093 | 0.335 | −3.948 | <0.001 | [0.195,0.577] |
| Dawn/dusk condition (1 if it is dawn/dusk, 0 otherwise) | 1.658 | 5.25 | 4.508 | <0.001 | [2.552,10.788] |
| *Accident type* | | | | | |
| Rear-end (1 if accident type is rear-end, 0 otherwise) | −0.783 | 0.457 | −4.186 | <0.001 | [0.317,0.659] |
| Broadside (1 if accident type is broadside, 0 otherwise) | −1.766 | 0.171 | −5.221 | <0.001 | [0.088,0.332] |
| *Accident outcomes* | | | | | |
| Moderate (1 if injury severity outcome is moderate, 0 otherwise) | −0.455 | 0.634 | −1.998 | 0.046 | [0.406,0.991] |
| Fatal (1 if injury severity outcome is fatal, 0 otherwise) | −2.335 | 0.097 | −8.733 | <0.001 | [0.057,0.164] |
| *Pre-accident conditions* | | | | | |
| Proceeding straight (1 if pre-accident movement is proceeding straight, 0 otherwise) | −1.210 | 0.299 | −4.544 | <0.001 | [0.178,0.504] |
| Run-off Road (1 if pre-accident movement is Run-off Road, 0 otherwise) | −3.88 | 0.021 | −9.545 | <0.001 | [0.009,0.046] |
| Entering traffic lane (1 if pre-accident movement is entering traffic lane, 0 otherwise) | −1.320 | 0.267 | −10.954 | <0.001 | [0.211,0.338] |
| Turn (1 if pre-accident movement is turn, 0 otherwise) | 0.687 | 1.988 | 4.012 | <0.001 | [1.421,2.781] |
| Backing (1 if pre-accident movement is backing, 0 otherwise) | −0.712 | 0.491 | −4.803 | <0.001 | [0.367,0.656] |
| Number of strata | 548 | | | | |
| McFadden pseudo-R-squared | 0.810 | | | | |
| Likelihood ratio test | 628.9 | | | | |
| Wald test | 344.2 | | | | |
| Score (log rank) test | 590.7 | | | | |

1 if accident is AV, 0 HDV; Sampling ratio is 1:5.
Sample size: 548 ADS accidents and HDV strata.

study. Control samples are accidents who do not have the specific outcome being studied. In this study, 'AV accident case sample' consists of AV accidents within each stratum, and the 'HDV accident control samples' consist of the HDV accidents within the same stratum. The method begins by utilizing an initial equal proportion of AV accident case samples to HDV accident control samples (1:1) and progressively increasing the ratio (1:3, 1:5, 1:7, 1:9...) until the coefficients between consecutive models exhibit no significant change. From Fig. 6, it is evident that there are no notable disparities between the models employing sample ratios of 1:5 and 1:6. As a result, we opt for the 1:5 ratio for our analysis. To further support our decision, we evaluate the improvement in log-likelihood across the models, aligning with our hypothesis of selecting an AV accident case sample to HDV-accident control sample ratio of 1:5.

As a result, 548 ADS accident accidents were applied for the matched case-control design and are discussed in this paper. The sample of the data is shown in Fig. 7. The estimation results and 95% confidence intervals of the odds ratio are presented in Table 1, which was generated using the survival package in R programming[88]. A total of 11 significant variables were identified by combining road and environment, accident type, accident outcomes, and pre-accident conditions during the estimation process.

## Data availability
The Human Driven-Vehicle (HDV) accidents dataset that we used to is publicly available at https://www.chp.ca.gov/programs-services/services-information/switrs-internet-statewide-integrated-traffic-records-system. The Autonomous Vehicle (AV) accidents dataset is available at https://github.com/UCF-SST-Lab/AVOID-Autonomous-Vehicle-Operation-Incident-Dataset/tree/main/asset. The Annual Average Daily Traffic (AADT) data of various road types from the California Traffic Census Program is available at https://dot.ca.gov/

programs/traffic-operations/census.Source data for figures are provided with this paper. All other data used in this study are available from the corresponding authors upon request. Source data are provided with this paper.

## Code availability

The codes for data validation and processing are available on Zenodo with a (https://doi.org/10.5281/zenodo.11081206). The quick tutorial and README file are also included in the repository for reference. Python scripts for geospatial data processing are prepared with the OSMnX package and offered in the repository, which can be referred to in the file Address2OSM.ipynb under the folder of code. All other code used in this study are available from the corresponding authors upon request.

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

## Acknowledgements

The authors wish to thank Dr. Ou Zheng for his role in creating the AVOID data used in this study.

## Author contributions

M.A.A. and S.D. conceived the study. M.A.A. and S.D. wrote the manuscript. M.A.A. and S.D. estimated the models and conducted the analysis. M.A.A. supervised the analysis and edited the manuscript.

## Competing interests

The authors declare no competing interests.
