## [Peer Review File · Nature Communications]

A matched case-control analysis of autonomous vs human-driven vehicle accidentsEditorial Note: Parts of this Peer Review File have been redacted as indicated to remove third-party material where no permission to publish could be obtained.

REVIEWER COMMENTS

Reviewer #1 (Remarks to the Author):

This is such an interesting read. Thank you very much to the authors for the effort. One of the major caveat is understanding how the data should be interpreted, especially given that this has not taken into account of no-accident in the same scenario. I understand the challenges in this, but perhaps it should somehow include the number of AVs and HDVs operating. I like the logic provided by the authors very much when it comes to explaining matched-case control study, which was 'involves comparing individuals who have a specific health outcome or disease (the cases) with individuals who do not have the health outcome or disease (the controls)'. Using the same logic, we should compare the chances of AV who involve in a specific road accident with AV who do not vs HDV. The discrepancy of the likelihood will be able to provide us with much more in-depth understanding. Please see below other comments from me:

- There are a lot of acronyms in this paper, it will improve readability if the authors explain them in full at least for the first time and also provide a definition (i.e., AV, HDV, ADS, ADAS...)
- Line 37-38: '..enriched with further parameters 598 ADS accidents in California'. Is the 598 refers to an increase of parameters or accidents?
- Line 54-55: 'being more cautious and following traffic laws more strictly' as compared to what/who?
- Line 55-56: 'the importance of understanding how AVs will interact with human drivers...'. This sentence does not follow the previous, as it is not a potential benefits of AVs.
- Line 64-87: these are all useful information but perhaps they can be grouped better to enhance readability, i.e., presenting findings that show AVs are safer first or vice versa, instead of having alternating arguments, because it does not seem like they are presented according to ascending order either (i.e., older to newer findings).
- Line 92: '..their automation level are shown in Figure 1'. When it comes to automation level, people often expects to see SAE levels of automation, but over here its split between ADAS and ADS. Perhaps again this is related to the first point above where more explanation/definition could be provided.
- Line 92-98: All these numbers/terms are very confusing and unclear. Also, how is this relate to line 37-38, where the author mentioned '598 ADS accidents'. The numbers in the paragraph and the figure does not match either. Authors also mentioned 2100 AV accidents was identified, but later mentioned that conventional modes of ADAS closely resemble HDV. So, is ADAS being considered as AV or HDV here?
- Line 105 to 107: the information on the pre-accident conditions and accident scene information for both CADMV and NHTSA accident data. Does the information related to these 2100 accidents or separate information? Please clarify.
- Line 122: 'according to the occurrence of AV accidents'. Is that base on year?
- Line 127: 'until the coefficients between consecutive models' exhibit no significant change. What factors are being taken into account of the matched case-control logistic regression model? What was the input and output?
- Figure 2: what are the y-axes?
- Again, it is very confusing when the number of data sets don't match. Line 131 and 132 mentioned that 'the ultimate dataset comprised 2740 HDV accidents as controls and 548 AV accidents as cases' but the 140-141 mentioned 'the full data set includes 2100 crashes' and then 'we utilize 35133 HDV accidents'.

- Given that table 1 focuses on binary outcome, 1 and 0, how useful these Mean and SD are? Wouldn't number of cases and percentages be more helpful?
- Thanks for explaining what Matched Case-Control Logistic Regression is in Section 4. Given that this has already been reported earlier, perhaps the explanation could come earlier. Moving the section up will be helpful.
- According to the logic in 159-161, cases should be AVs/HDVs involved in accidents, and controls would be AVs/HDVs not involved in accidents? But this follows correctly in 166-167, whereby it is to investigate the impact of various factors on the likelihood of accidents in two specific scenarios. Did the authors also look at no accidents situation? Or else how do we know about the likelihood of accidents in a given scenario? Please clarify if I understood incorrectly.
- Table 1, can the authors please clarify the motivation behind the 3 categories? AV (ADS+ADAS); ADS (for matched case control study) and HDV? I believe that was the reason of all the different numbers of cases presented above.
- Results line 197-198: I do believe that the AV (2100) here consists of both SAE level 4 ADS (1099) and level 2 ADAS (1001). Instead of 2100+1099+1001. It will be good to clarify. Also finally seeing the SAE levels which most will be more familiar with, perhaps this should already be included or clarified in the beginning.
- Line 200-201: HDVs have a slightly higher occurrence rate at 83% compared to 77% for AVs. Comment: how does the number of HDVs vs AVs taken into account? -- same for all the other interpretation. How should the reader interpret this finding actually? Because I don't think we can say that if there is an accident during day time, there is 83% chance of it being a HDV vs 77% chance for AVs, or can we? Or perhaps, if there is a HDV accident, there is an 83% chance that it occurs during day time, but x % occur other conditions? Same for the whole results section.
- Line 205-208: this finding is indeed interesting, perhaps include some descriptive information here.
- Line 217-221, what are the factors that are included in the model but not significant?
- Line 221: 'This indicates a higher probability of an AV accident occurrence when these variables are present': Comment: what do 'these variables are present' means, what is the direction? Do you mean that during dawn/dusk, and when turn indicator are present on the other vehicle, AV accident occurrence are higher? This is rather strange, do you know whether the turn indicator provided by the other vehicles are valid?
- Line 224-226: I realised that the authors use 'indicator' as 'factor' perhaps they are redundant because they all are factors. Perhaps the comment above 'turn indicator' refers to a turning situation of AV (or other vehicle) rather than the turn signals? Anyway, There are a lot of terms here that I do not understand (i.e., moderate severity, proceeding straight indicator, run-off road, backing indicator)
- Figure 3 to 6, it will be useful to provide (N = and % information to each category)
- Line 249-150: Perhaps it is due to the help of advanced algorithms, but do the authors know what is the chances of HDV hitting HDV? If HDV hitting AV is significantly higher than HDV hitting HDV, it could also be interpreted as AV was not designed in a way that can be understood and predicted by the human driver following behind (I.e., abrupt stopping with no valid reason and in weird places?)
- Line 252. I do not follow: 'when in conventional mode, half of the AV vehicles are moving, and the others are stopping'.
- Line 257-258: in terms of accident severity, 82% of accidents occur as minor when HDV hit AV

(does that mean 18% is severe?). This percentage is 67% when the AV hit HDV (does that mean 67% as minor but 33% as severe?). It seems to contradict the above that says HDV accidents are more severe?

- Figure 7: please include % in the figure. Also, it is perhaps important to clarify why these AVs were operating in conventional mode? Does it indicate that those are the more complex situation where AV cannot handle? And thus the higher rate of accident in AV hit HDV driven in conventional mode could have been a bit of an unfair comparison? Worth discussing.
- Please clarify that the authors first make sure that the accidents happened in the same location (or within 5 miles radius), same day of the week and time of day, before matching with the other variables such as those mentioned in the table 2? Was traffic patterns assumed based on the same day of the week and time of day?
- Line 289: 'This indicates a lower likelihood of an AV accident in rainy weather compared to an HDV accident'. I appreciate that the authors mentioned that these vehicles are seldom tested in adverse weather. Therefore it pretty much goes back to the previous question on the 'accident vs no accident ratio' under each condition for these vehicle type. Understanding the discrepancy would provide a better understanding on the likelihood.
- The line in 300 to 302 quickly contradicts with 297-298. Authors should be very careful in making these claims on sensor capabilities, although I understand citations were provided. But how accurate these really are? Not only in terms of detection but intention recognition? especially followed by the discussion on Situation Awareness below.
- Line 316 to 327: interesting, perhaps expand what situation awareness is and provide a citation, but this explanation is rather vague, do the authors know why is turning somehow 'more difficult' than other situation?

Reviewer #2 (Remarks to the Author):

Summary of paper content

The work is analyzing accident data of autonomous vehicles statistically and compares them with accident data of human driven vehicles. It uses a matched case-control analysis in which it uses the AV data as cases and the HDV data as controls. It tries to derive in which situations AV accidents are more (less) likely to happen than HDV accidents.

Novelty & Relevance

The topic to analyze accidents of autonomous vehicles is highly relevant to understand the potentials of autonomous driving in comparison with human driving. Therefore, the paper discusses a very interesting question that isn't yet analyzed sufficiently well in previous work. The results provide some novel scientific insights.

Organisation & Readability

The organisation and readability of the paper in general is good. The level of English is good.

Some minor points are:

line 47: please introduce the full name of the NHTSA before using the abbreviation. I guess you meant the National Highway Traffic Safety Administration but that might not be clear to all readers.

line 171: please add an 's' at the end of the word "depend"

line 174: I am not sure whether "gauge" is the perfect word here. You might like to replace it by another expression like "estimate" or "derive"

line 243: what does the word "ed" mean?

Technical Soundness

Section 3.1: It would be interesting to get more information on the data. Which scenarios do we find in the data, highways, rural roads, urban driving? Which collision types do they contain to which extend?

Section 3.2, Table 1: I would have expected some more criteria being part of the list, especially (i) vehicle speed, and (ii) right-of-way at intersections (i.e. yield/stop/priority/traffic lights). Why didn't you consider these?

Section 4: The description of your matched case-control logistic regression model was very unclear to me. I could not fully follow the description so I will go through the text and tell you what I understood and what looked strange to me.

Lines 159-165: seems to be clear to me. However, I am wondering whether the application of using AV accidents as cases and HDV accidents as controls really is appropriate. In references 23 and 24 accidents are used as cases and non-accidents as controls and the idea is to find out the most relevant factors to explain the occurrence of accidents. Here, you deal with accident cases only and compare AV accidents with HDV accidents. This will not allow you to find out which factors are relevant for the occurrence of accidents in general but only allows you to compare the distributions of AV accidents and HDV accidents over various situations. I doubt that you can draw the conclusions that you claim in Section 5 since you completely ignore the non-accident cases and so you ignore the prior probability of accidents of AVs and HDVs (i.e. the number of accidents per distance traveled).

Line 169: I looked in reference 25 but I did not understand why it is cited here.

Line 170: Are "small k" and "capital K" the same variable or different variables? I assume it should be the same.

Line 170: You mention that one stratum contains k HDV accident samples. How do you use the word "sample" here? I would interpret a sample as a set of accidents which would mean that one stratum consists of one AV accident and k sets of HDV accidents. Or did you mean "sample element" rather than "sample" (i.e. a single HDV accident)? Then, it would make more sense. Please check your wording.

Line 171-172: You use the word "observation" and the expression "the number of accidents recorded in this stratum". If I assume the second interpretation of line 170 (see above), the number of accidents recorded in this stratum is k+1. So what is the number of observations? I would guess, it is the same, i.e. k+1. But why do you use two different expressions for the same? That remains unclear to me.

Line 172-175: You introduce $P_{ri}(X_{ji})$ as the probability that the j-th observation in the i-th stratum is an accident. According to what was said before all observations are accidents. So $P_{ri}(X_{ji})=1$. Or are there some non-accidents in the data set? Or should $P_{ri}(X_{ji})$ be the probability that the j-th observation in the i-th stratum is an AV accident? Please clarify it.

Line 172-175: First, you are using X_{ji} , later on you are using X_{ij} . Why do you exchange the indexes?

Line 182: are "small x_{uji} " in (2) the same as "capital X_{uji} " in (1) or not?

By the way, the whole mathematical notation in this chapter is rather sloppy and confusing. A strict distinction between probabilities, frequencies, random events, random variables, estimates, and other terms would be rather helpful to improve readability. In its present form I don't feel that the notation is sufficient for publication. E.g. the term $P_{ri}(X_{ji})$ would indicate that X_{ji} is a random event since probabilities are only defined for random events. It might also be a random variable (the choice of a capital letter would indicate this) in a sloppy mathematical notation (which, however, you find rather often in literature), however, it is hard to assess how the term would look like in strict mathematical notation. The term $X_{ij}=(X_{1ij}, \dots)$ however would indicate that X_{ij} (or X_{ji} ?) is just a data point and neither a random variable nor a random event.

So I guess that the whole Chapter 4 requires careful rewriting.

Section 5 (and 6): As mentioned above, I doubt that you can conclude from your study that AV accidents are more (or less) likely in some situations than HDV accidents since you completely ignore the prior probabilities of accidents. The only conclusions that you may draw are that the distribution of AV accidents over various situations differs from the distribution of HDV accidents. I think, this point should be made clearer. Otherwise, the reader gets an erroneous impression.

Section 5 Figure 7: I was unable to interpret this figure. Please explain it in more detail.

Section 5.3 & 5.4: Some of your explanations are debatable or need refinement from a technical point of view.

* line 293-294: AVs cannot react within milliseconds, at least not within a few milliseconds.

Sensing, computing, and acting requires time. For all technical systems that I know the reaction time is a couple of 100 milliseconds.

* line 295-298: Bad weather conditions affect the perception abilities of cameras a lot, of lidar systems to some extent. Only radar systems and GNSS do not suffer that much from bad weather but they have other shortcomings.

* line 308-314: The explanation that AVs avoid rear-end accidents because they react faster is possible but not certain. Another explanation might be that AVs keep larger distances than HDVs in dense traffic. Without knowing more details about these accidents we cannot decide which explanation fits.

* line 319-321: Limited sensor ranges and limited coverage of the environment by sensors might also be an explanation.

* line 322-323: Different AVs follow different driving strategies. Some might follow predefined strategies but there are also vehicles which calculate their behavior as reaction to the present situation so that they are able to deal with unseen situations.

* line 324-327: Human drivers might have driven several hundred thousands of kilometers themselves. AVs are tested on millions or even more kilometers. Who of both has more driving experience?

* line 358-359: Please replace "reasons are" by "reasons might be" since your analysis does not reveal the technical reasons for your observations and your explanations are just hypotheses.

Self-assessment and research background of the reviewer

The reviewer has >15 years of experience in autonomous driving and >20 years of experience in machine learning. The reviewer's background is computer science with strong focus on math (including probability theory). The reviewer is familiar with all technical aspects of autonomous driving and current developments in this field.

A Matched Case-Control Analysis of Autonomous vs Human-Driven Vehicle Accidents

Modifications and Responses to the Reviewers' Comments and Suggestions

The authors are grateful to the reviewers for their careful review and feedback. The manuscript was revised to address all the Reviewers' comments. For your convenience added or modified parts are marked **in blue** in both the Response document and the Revised Manuscript.

Reviewer #1

General Comment:

This is such an interesting read. Thank you very much to the authors for the effort. One of the major caveat is understanding how the data should be interpreted, especially given that this has not taken into account of no-accident in the same scenario. I understand the challenges in this, but perhaps it should somehow include the number of AVs and HDVs operating. I like the logic provided by the authors very much when it comes to explaining matched-case control study, which was ‘involves comparing individuals who have a specific health outcome or disease (the cases) with individuals who do not have the health outcome or disease (the controls)’. Using the same logic, we should compare the chances of AV who involve in a specific road accident with AV who do not vs HDV. The discrepancy of the likelihood will be able to provide us with much more in-depth understanding.

Response:

We sincerely appreciate your thoughtful comments and the positive feedback on our work. Your insights and suggestions are invaluable in enhancing the quality and comprehensiveness of our research.

For the main concern about the inclusion of no-accident scenarios, on one hand, we sourced data on autonomous vehicle miles traveled (AVMT), disengagement accidents, and accident occurrences from the publicly available California DMV AV accident and disengagement reports. For the HDV group, we gathered data on Motor Vehicle Miles of Travel (MVMT) from the SWITRS Annual Report of Fatal and Injury Motor Vehicle Traffic Collisions for non-accident analysis. On the other hand, we also incorporated Annual Average Daily Traffic (AADT) data at specific accident sites from the California Traffic Census Program. This enabled us to calculate the accident rate under different traffic conditions.

For the concern about comparing the data, our aim is to explore the differential characteristics of AV and HDV accidents. The distribution of AV accidents over various conditions differs from the HDV accidents can be concluded from the matched case-control study (AV accidents as cases and HDV accidents as controls) to interpret the data. Attempting to use non-crashes in matched case-control study might not be feasible since most of the factors investigated would not exist in the control cases.

Please also note that the AV vehicles and therefore their crashes are extremely small percentage and thus any sample of AV accidents vs no accidents would be extremely unbalanced prohibiting reaching any meaningful results.

Comment 1:

There are a lot of acronyms in this paper, it will improve readability if the authors explain them in full at least for the first time and also provide a definition (i.e., AV, HDV, ADS, ADAS...).

Response:

In response to your suggestion, we revised the manuscript to include full explanations of each acronym upon its first use. Specifically:

Line 25: Autonomous Vehicles (AV)

Line 36: Annual Miles Traveled by Autonomous Vehicles (AVMT)

Line 38: California Department of Motor Vehicles (CADMV)

Line 39: Advanced Driving Systems (ADS) SAE Level 4

Line 44: Human-driven Vehicles (HDV)

Line 50: National Highway Traffic Safety Administration (NHTSA)

Line 51: Advanced Driver Assistance Systems (ADAS) SAE Level 2

Comment 2:

Line 37-38: '..enriched with further parameters 598 ADS accidents in California'. Is the 598 refers to an increase of parameters or accidents?

Response:

The number 598 refers specifically to the total count of ADS accidents that occurred in California, not to an increase in parameters. The phrase "enriched with further parameters" is meant to indicate that our dataset includes additional details about each of these accidents by imputing missing data using Open Weather API and Google Map API. We apologize for any confusion caused by the wording and clarified this in the manuscript:

Line 39-40:

Up until June 2023, we have identified and enriched with additional factors 598 ADS accidents in California (AVOID dataset)⁶.

Comment 3:

Line 54-55: 'being more cautious and following traffic laws more strictly' as compared to what/who?

Response:

The phrase 'being more cautious and following traffic laws more strictly' refers to a comparison with “some” **human drivers**. The revised manuscript now includes a direct comparison with human drivers for clarity:

Line 66-69:

Zhou et al. (2019) explored the behavior of Adaptive cruise control (ACC) system for speed adjustment and found out that AVs may exhibit some human-like behaviors ¹¹, and being more cautious and following traffic laws more strictly than human drivers in general when autonomous overtaking function is engaged ¹².

Comment 4:

Line 55-56: 'the importance of understanding how AVs will interact with human drivers...'. This sentence does not follow the previous, as it is not a potential benefit of AVs.

Response:

To address your concern, we have reorganized and revised this section and deleted this sentence. This paragraph introduces the benefits of ADAS (SAE Level 2), ADS (SAE Level 4), advantages of AV over HDV, then summarizes the possible safety challenges of AVs. The updated text is as follows:

Potential benefit of ADAS:

Line 57-60:

ADAS functions including Electronic Stability Control (ESC) and Anti-Lock Braking (ABS), help avoid accidents in critical situations, while Information and Communication Technology (ICT) aids in ongoing driving tasks to prevent accidents. For example, Tesla Autopilot consistently maintains a closer distance to the lane center than human drivers ⁷.

Potential benefit of ADS:

Line 60-63:

Additionally, Level 3 or higher levels of automation can further enhance traffic by reducing crashes¹, improving mobility for the disabled and elderly, and minimizing traffic collisions through efficient driving and the reduction of human errors⁸.

Advantages of AV over HDV:

Line 63-69:

Koopman and Wagner (2017) highlighted the importance of understanding how AVs will interact with human drivers, pedestrians, and other road users ⁹. Compared with HDV, accident avoidance features of AVs can significantly reduce accidents and fatalities caused by distracted driving or human error by helping control the vehicle and alerting drivers to potential dangers ¹⁰. Zhou et al. (2019) explored the behavior of Adaptive Cruise Control (ACC) system for speed adjustment and found out that AVs may exhibit some human-like behaviors ¹¹,

and being more cautious and following traffic laws more strictly than human drivers in general when autonomous overtaking function is engaged¹².

Comment 5:

Line 64-87: these are all useful information but perhaps they can be grouped better to enhance readability, i.e., presenting findings that show AVs are safer first or vice versa, instead of having alternating arguments, because it does not seem like they are presented according to ascending order either (i.e., older to newer findings).

Response:

In response to your insightful comment, we have reorganized the section for improved readability. The revised manuscript now arranges the findings in a structured way, examining the topic from two angles: first, how AV safety surpasses HDV, and second, the advantages of HDV over AV. The structure of the manuscript progresses from earlier to more recent studies:

Line 83-99:

The comparison of safety performance between AV and HDV is a topic of debate, with conflicting viewpoints¹⁶. On one side, numerous studies support the view that AVs are generally safer than HDVs. For instance, Dixit et al. (2016) analyzed the statistical distribution of reports on 69 manual disengagements and accidents, they compared the accident rates of Google self-driving cars to those of human drivers and found that fatal accidents involving Google cars were lower than that of HDV, no fatalities occurred as compared to 1 death for every 108 million miles in California¹⁷. Additionally, from 2009 to 2015 in Mountain View, California, Google cars demonstrated a significantly lower rate of police-reportable accidents per million vehicle miles traveled (VMT) compared to human drivers (2.19 versus 6.06)¹⁸. On the other hand, some research challenge this view, suggesting that the safety of AVs may not always exceed that of HDVs. Schoettle and Sivak (2015) uncovered that AVs have a higher rate of accidents per million miles traveled compared to HDVs in limited (and generally less demanding) conditions (e.g., avoiding snowy areas) with 11 accident records¹⁹. Favarò et al. (2017) found that rear-end accidents were the most frequent type of AV accidents, with AVs being hit from behind by conventional vehicles at a rate twice that of rear-end accidents than rear-end “fender-benders” for conventional vehicles in California 2013²⁰. Shetty et al. (2021) found that the type of AV technology used was a significant predictor of accident risk, with vehicles that were less autonomous (i.e., requiring more human intervention) having a higher risk of accidents under different scenarios²¹.

Comment 6:

Line 92: '..their automation level are shown in Figure 1'. When it comes to automation level, people often expects to see SAE levels of automation, but over here its split between ADAS and ADS. Perhaps again this is related to the first point above where more explanation/definition could be provided.

Response:

To clarify, in our study, we distinguish between Advanced Driver Assistance Systems (ADAS) and Advanced Driving Systems (ADS) based on their respective levels of automation as defined by the Society of Automotive Engineers (SAE). Specifically, ADAS corresponds to SAE level 2, which involves partial automation, whereas ADS is aligned with SAE level 4, indicating high automation. We have updated Figure 1 to better illustrate these distinctions and provide a clearer understanding of the different levels of automation within the context of our research.

Figure 1. Number of accidents related to the studies of vehicles with driving automation (ADS / ADAS)

Comment 7:

Line 92-98: All these numbers/terms are very confusing and unclear. Also, how is this relate to line 37-38, where the author mentioned '598 ADS accidents'. The numbers in the paragraph and the figure does not match either. Authors also mentioned 2100 AV accidents was identified, but later mentioned that conventional modes of ADAS closely resemble HDV. So, is ADAS being considered as AV or HDV here?

Response:

The figure below illustrates the data mentioned. The 598 ADS accidents are the initial data collected from the California Department of Motor Vehicles (CA DMV), which forms a part of the AV accident data for analyzing general trends. Out of these, only 548 ADS accidents were suitable for the matched case-control study. The reduction in number is due to missing data or instances where matching with HDV data was not feasible. The detailed description of the data applied in this paper is illustrated in the chart below:

Figure 2. Sample size applied in this research.

For the general trend analysis, both ADS and ADAS are included under the AV category because we are interested in the overall impact of autonomous technologies on vehicle safety. ADAS-equipped vehicles, although not fully autonomous, incorporate significant automation features that can influence driving patterns and safety outcomes.

For the matched case-control study, the focus is narrower. This part aims to assess the safety performance of ADS in a controlled manner, comparing them directly to HDV. ADS-equipped vehicles operate with higher levels of automation, where the vehicle can drive under limited conditions and will not operate unless all required conditions are met. However, even with the most advanced ADAS, the human driver is still responsible for monitoring the driving environment and is expected to remain engaged with the driving task. The distinction in the matched case-control study is to ensure that the control group (HDV) is distinctly separate from the case group (ADS) in terms of vehicle control and operation.

Please note ADAS are not combined with HDV or considered HDV. We mentioned that “the manual override and conventional modes of ADAS closely resemble HDV” as part of the discussion about the Level 2 (ADAS) as they are still controlled by humans and operate when the equipment is disengaged as HDV”.

In this paper we not only focused on the matched case control of ADS level 4 vs HDV with an AV sample of 548, but we also provided other information such as the various factors for provided HDV, ADAS, ADS and ADAS+ADS, so the reader can contrast any to the others. Of course, in each the sample differed (Figure 2, Figures 6-9).

Comment 8:

Line 105 to 107: the information on the pre-accident conditions and accident scene information for both CADMV and NHTSA accident data. Does the information relate to these 2100 accidents or separate information? Please clarify.

Response:

In the manuscript, the 2100 accidents encompass pre-accident conditions and accident scene details, sourced from both the California Department of Motor Vehicles (CADMV) and the National Highway Traffic Safety Administration (NHTSA). Please refer to Figure 2, 2100 are all ADAS + ADS accidents from both databases, before any enrichment or screening. To clarify further in the paper, it is added as follows:

Line 157-158:

The 2100 accidents include variables of pre-accident condition and accident scene information from both CADMV and NHTSA.

Comment 9:

Line 122: 'according to the occurrence of AV accidents'. Is that based on year?

Response:

To clarify, the data collection for both AV and HDV accidents occurred concurrently over the same time period based on the day of the week and the time of day. This strategy was critical to reduce the influence of confounding factors on the association between risk elements and traffic accident outcomes. In our study, we meticulously matched the cases (AV accidents) with the controls (HDV accidents) in identical locations. In instances where a location had insufficient control data, we extended our search to encompass similar areas within a 5-mile radius. Moreover, we considered road types in each stratum to ensure similar geometric characteristics. It is clarified in the manuscript:

Line 322-326:

To overcome this challenge of variables that confound the relationship between risk factors and traffic accident outcomes, the first principle is to match cases and controls at the same location. In the case of a location that does not have enough controls, similar locations within a radius of 5 miles were used, and the day of the week and time of day were controlled. We assumed that cases and controls were under similar traffic patterns based on the controlled time and space.

Comment 10:

Line 127: 'until the coefficients between consecutive models' exhibit no significant change. What factors are being taken into account of the matched case-control logistic regression model? What was the input and output?

Response:

As detailed in Table 2 of the manuscript, the input variables for the model encompass a range of road, environmental conditions, accident types, accident outcomes, and pre-accident conditions.

The output would be the likelihood of an ADS accident (and Odds ratios), That includes the estimated parameters for these variables, which are then used to compute the odds ratios, t-statistics, p-values, and 95% confidence intervals of the odds ratios.

(Factors considered for matched case-control study)	Estimated parameter	Odds Ratio	t-statistic	P value	95% Confidence Interval of OR
Road and environment					
Rain (1 if weather is rain, 0 otherwise)	-1.098	0.334	-8.038	<0.001	[0.255,0.436]
Dawn/dusk (1 if it is dawn/dusk, 0 otherwise)	1.639	5.15	8.962	<0.001	[3.599,7.370]
Accident type					
Rear-end (1 if accident type is rear-end, 0 otherwise)	-0.891	0.41	-3.815	<0.001	[0.260,0.649]
Broadside (1 if accident type is broadside, 0 otherwise)	-1.662	0.189	-2.167	0.03	[0.042,0.853]
Accident outcomes					
Moderate (1 if injury severity outcome is moderate, 0 otherwise)	-0.496	0.609	-2.773	0.006	[0.429,0.865]
Fatal (1 if injury severity outcome is fatal, 0 otherwise)	-0.525	0.592	-2.09	0.037	[0.362,0.968]

Pre-accident conditions					
Proceeding straight (1 if pre-accident movement is proceeding straight, 0 otherwise)	-0.831	0.436	-6.977	<0.001	[0.345,0.550]
Run-off Road (1 if pre-accident movement is Run-off Road, 0 otherwise)	-1.126	0.325	-2.918	0.004	[0.152,0.691]
Entering traffic lane (1 if pre-accident movement is entering traffic lane, 0 otherwise)	-1.746	0.175	-5.39	<0.001	[0.093,0.329]
Turn (1 if pre-accident movement is turn, 0 otherwise)	0.317	1.373	2.036	0.042	[1.012,1.864]
Backing (1 if pre-accident movement is backing, 0 otherwise)	-0.63	0.533	-2.42	0.016	[0.350,0.887]
	Number of strata				548
	McFadden pseudo-R-squared				0.602
	Likelihood ratio test				50.58
	Wald test				46.95
	Score (log rank) test				48.37

Comment 11:

Figure 2: what are the y-axes?

Response:

On the left side of the y-axis, we display the average changes in the regression coefficients. On the right side of the y-axis, the changes in log-likelihood values are plotted. We have revised the Figure to include these labels on the y-axes for better clarity.

Figure 4. Selection for the optimal number of controls

Comment 12:

Again, it is very confusing when the number of data sets don't match. Line 131 and 132 mentioned that 'the ultimate dataset comprised 2740 HDV accidents as controls and 548 AV accidents as cases' but the 140-141 mentioned 'the full data set includes 2100 crashes' and then 'we utilize 35133 HDV accidents'.

Response:

We acknowledge that the different samples might be confusing. We assembled large and diverse samples to show the readers the maximum available AV data from all levels and sources. We hope Figure 2 has organized and demonstrated the different sub datasets. As depicted in the chart below, for the matched case-control study, we utilized data of 548 ADS accidents and 2740 accidents HDV.

For the general trends analysis encompassed a broader dataset, including 35,133 HDV accidents and 2100 AV accidents involving both ADAS and ADS.

The clarification of data is shown as the chart below:

Figure 2. Sample size applied in this research.

Comment 13:

Given that table 1 focuses on binary outcome, 1 and 0, how useful these Mean and SD are? Would number of cases and percentages be more helpful?

Response:

The mean in the context of binary variables, effectively serves as an indication of the percentage of data that have the value 1. For instance, a mean value of 0.77 for the 'Clear' variable implies that clear sky conditions were present in 77% of the cases.

The standard deviation in this case measures the variability or spread of the binary data. A lower standard deviation indicates that the outcomes are more consistently near the mean, meaning that there is a higher prevalence of either 0 or 1, depending on which one is closer to the mean. A higher standard deviation in a binary variable indicates that the data points are more balanced in terms of occurrences of 1s and 0s, with neither being overwhelmingly more common than the other.

The meaning of the mean as a percentage and standard deviation has been incorporated as note under Table 1:

Table 1. Summary statistics for the variables included in the generated datasets.

Variable type	Variable description	AV (ADS+ADAS) (2100)		ADS (For matched case control study) (548)		HDV (35,133)	
		Mean	Standard deviation	Mean	Standard deviation	Mean	Standard Deviation
(.....)							

*Note:

The mean serves as an indication of the percentage of data that have the value 1.

The standard deviation measures the variability of the binary data. A lower standard deviation suggests results near the mean, indicating a dominance of either 0 or 1, while a higher standard deviation indicates a more balanced distribution of 0 and 1.

Comment 14:

Thanks for explaining what Matched Case-Control Logistic Regression is in Section 4. Given that this has already been reported earlier, perhaps the explanation could come earlier. Moving the section up will be helpful.

Response:

We appreciate your suggestion to reposition the explanation of Matched Case-Control Logistic Regression for improved clarity. Following your recommendation, we have advanced this section to precede the data preparation portion of the manuscript (now section 3).

Comment 15:

According to the logic in 159-161, cases should be AVs/HDVs involved in accidents, and controls would be AVs/HDVs not involved in accidents? But this follows correctly in 166-167, whereby it is to investigate the impact of various factors on the likelihood of accidents in two specific scenarios. Did the authors also looked at no accidents situation? Or else how do we know about the likelihood of accidents in a given scenario? Please clarify if I understood incorrectly.

Response:

The cases are AVs involved in accidents, and the controls are HDVs in accidents. This design is to investigate the difference of related factors in the two groups of accidents (AV and HDV).

To delve deeper into the accident likelihood under non-accident conditions for AVs, we sourced data on autonomous vehicle miles traveled (AVMT), disengagement accidents, and accident occurrences from the publicly available California DMV AV accident and disengagement reports. We calculated the disengagement rate per 1,000 miles and the accident rate per 1,000 miles.

Number of AV AVMT, disengagement and Accidents

Year	AVMT	Disengagement	Disengagement/1000 mi	Accident	Accident /1000 mi
2017	483786	11281	23.32	22	0.05
2018	1971474	66049	33.5	44	0.02
2019	2636198	1972	0.75	67	0.03
2020	1565353	335	0.21	20	0.01
2021	4051850	8216	0.2	117	0.03
2022	5964804	2376	0.04	150	0.03

Number of AV AVMT and Accidents

For the HDV group, we gathered data on Motor Vehicle Miles of Travel (MVMT) from the SWITRS Annual Report of Fatal and Injury Motor Vehicle Traffic Collisions. This information allowed us to compute the accident rate per 1 million miles for HDVs.

Number of HDV MVMT and Accidents

Year	MVMT	Accident	Accident /1 million mi
2010	327770000000	161094	0.05
2011	325032000000	159115	0.05
2012	326547000000	159696	0.05
2013	329174000000	156909	0.05
2014	334664000000	162742	0.05
2015	339843000000	178669	0.05
2016	342853000000	195347	0.06
2017	344304000000	193564	0.06
2018	347194000000	191971	0.06
2019	351151000000	187211	0.05

*Data after 2019 is Not yet available

To be noticed, these Accident rates based on miles traveled offer only a general comparison. This metric does not directly reveal the distinct features or allow for a direct comparison of AV and HDV accidents, particularly in terms of different types of accidents at specific locations. Direct comparison between AV and HDV crashes is still not viable as the difference in exposure and number of vehicles of both types are extremely unbalanced. To address this, we incorporated Annual Average Daily Traffic (AADT) data at specific accident sites from the California Traffic Census Program. This enabled us to calculate the accident rate under different traffic conditions.

The method for calculating accident rates takes into consideration vehicle exposure by including AADT. The accident rates at segments can be determined using the equation below:

$$R_{section} = \frac{A * 1,000,000}{365 * T * V_{section} * L}$$

Where

$R_{section}$ is the accident rate for the section.

A is the number of reported accidents during the time period.

T is the number of years considered.

$V_{section}$ is AADT or annual average daily traffic, vehicles per day.

L is the length of section, miles.

For intersections, we use the total entering AADT as the variable V. The accident rates at intersections can be determined using the equation below:

$$R_{spot} = \frac{A * 1,000,000}{365 * T * V_{spot}}$$

Where

R_{spot} is the accident rate for the spot.

A is the number of reported accidents during the time period.

T is the number of years considered.

V_{spot} is AADT or annual average daily traffic ($V_{spot} = (AADT_{minor\ roads} + AADT_{major\ roads})$), vehicles per day.

The accident rate and number of accident comparison by road type is shown in the Figure below. Accident rates are classified into four categories: Highways, Intersections, Rural Roads, and Streets. HDVs show a higher incidence of accidents on highways, intersections, and streets, particularly on highways. For rural roads, HDVs and AVs exhibit almost similar accident rates. Across all road types, HDVs consistently record significantly higher accident figures than AVs.

Figure. Comparison of Accident Rate

This part is summarized as below in the manuscript:

Line 329-334:

Direct comparison between AV and HDV crashes might not be possible, as the difference in exposure and number of vehicles of both types are extremely unbalanced. To address this, we also incorporated Annual Average Daily Traffic (AADT) data of various road types from the California Traffic Census Program. HDVs show a higher incidence of accidents on highways, intersections, and streets, particularly on highways. For rural roads, HDVs and AVs exhibit almost similar accident rates. Across all road types, HDVs consistently record significantly higher accident rates than AVs.

Among the novelty of using matched case design was to be able to validly compare AV vs HDV within strata.

Comment 16:

Table 1, can the authors please clarify the motivation behind the 3 categories? AV (ADS+ADAS); ADS (for matched case control study) and HDV? I believe that was the reason of all the different numbers of cases presented above.

Response:

As depicted in the chart below, for the matched case-control study, we utilized data from 548 ADS accidents (ADS (for matched case control study) in Table 1).

For the general trends analysis encompassed a broader dataset, including 35,133 HDV accidents (HDV in Table 1) and 2100 AV accidents involving both ADAS and ADS accidents (AV (ADS+ADAS) in Table 1).

For the general trend analysis, both ADS and ADAS are included under the AV category because we are interested in the overall impact of autonomous technologies on vehicle safety. ADAS-equipped vehicles, although not fully autonomous, incorporate significant automation features that can influence driving patterns and safety outcomes. For the matched case-control study, the focus is narrower. This part aims to assess the safety performance of ADS in a controlled manner, comparing them directly to HDV. ADS-equipped vehicles operate with higher levels of automation, where the vehicle can drive under limited conditions and will not operate unless all required conditions are met. However, even with the most advanced ADAS, the human driver is still responsible for monitoring the driving environment and is expected to remain engaged with the driving task. The distinction in the matched case-control study is to ensure that the control group (HDV) is distinctly separate from the case group (ADS) in terms of vehicle control and operation.

The clarification of data is shown as the chart below:

The number of samples is also added in Table 1:

Variable type	Variable description	AV (ADS+ADAS) (2100)	ADS (For matched case control study) (548)	HDV (35,133)
---------------	----------------------	----------------------	--	--------------

Comment 17:

Results line 197-198: I do believe that the AV (2100) here consists of both SAE level 4 ADS (1099) and level 2 ADAS (1001). Instead of 2100+1099+1001. It will be good to clarify. Also finally seeing the SAE levels which most will be more familiar with, perhaps this should already be included or clarified in the beginning.

Response:

We have clarified that the AV group, which comprises 2,100 accidents, includes both 1,099 accidents involving vehicles with SAE level 4 Advanced Driving Systems (ADS) and 1,001 accidents involving vehicles with SAE level 2 Advanced Driver-Assistance Systems (ADAS). The total of 2,100 accidents represents the combined data for these two levels of vehicle automation, not an additive figure. It is clarified that in the manuscript:

Line 240-241:

Figures 6 (a,b,c,d) display the distribution of factors affecting AV (2,100), including SAE level 4 ADS (1099), level 2 ADAS (1001) and HDV accidents (35,133), respectively.

Moreover, we have taken your advice and included a clarification on SAE levels at the beginning of the document to ensure that readers are familiar with these classifications in the introduction:

Line 51: Advanced Driver Assistance Systems (ADAS) SAE level 2

Line 39: Advanced Driving Systems (ADS) SAE level 4

Comment 18:

Line 200-201: HDVs have a slightly higher occurrence rate at 83% compared to 77% for AVs. Comment: how does the number of HDVs vs AVs taken into account? -- same for all the other interpretation. How should the reader interpret this finding actually? Because I don't think we can say that if there is an accident during day time, there is 83% chance of it being a HDV vs 77% chance for AVs, or can we? Or perhaps, if there is a HDV accident, there is an 83% chance that it occurs during daytime, but x % occur other conditions? Same for the whole results section.

Response:

It is important to clarify that these percentages do not imply the probability of an accident type occurring during clear conditions, but rather the proportion of each accident type that occurs in such weather. If we look at all HDV accidents, 83% took place under clear skies, not that there is an 83% chance of any given accident being an HDV. Similarly, for AV accidents, 73% occurred during clear weather conditions.

Line 242-247:

Evaluating environmental factors, the majority of accidents involving both AVs and HDVs tend to happen under clear weather conditions. Notably, accidents involving HDVs occur slightly more frequently under these conditions, at a rate of 83%, compared to 73% for AVs. However, AVs are more commonly involved in accidents during rainy or foggy conditions, accounting for 12% of such accidents, compared to HDVs which experience these conditions in only 5% of accidents.

Comment 19:

Line 205-208: this finding is indeed interesting, perhaps include some descriptive information here.

Response:

We acknowledge your suggestion to enhance the discussion of our findings with descriptive details. Accordingly, we have expanded the section in question to include descriptive information that encompasses several key aspects: Road and Environment, Pre-accident Conditions, Accident Type, and Accident Outcomes.

Road and environment:

Line 242-247:

Evaluating environmental factors, the majority of accidents involving both AVs and HDVs tend to happen under clear weather conditions. Notably, accidents involving HDVs occur slightly more frequently under these conditions, at a rate of 83%, compared to 73% for AVs. However, AVs are more commonly involved in accidents during rainy or foggy conditions, accounting for 12% of such accidents, compared to HDVs which experience these conditions in only 5% of accidents. About 5% of AV accidents take place in locations impacted by previous traffic events or work zones, where normal traffic is disrupted by earlier events like accidents, disabled vehicles, or spilled cargo. In comparison, just 1.3% of HDV accidents occur in similar settings.

Accident type:

Line 249-251:

In terms of accident type, rear-end accidents constitute a majority for both AVs and HDVs, accounting for 39% and 45% of accidents, respectively.

Pre-accident conditions:

Line 251-255:

For both AVs and HDVs, the most frequent pre-accident movement is proceeding straight. It is observed that 56% of AV accidents and 58% of HDV accidents occur under this specific condition. Notably, other vehicles make up 80% of involvements in AV accidents, with pedestrians accounting for 3%. In contrast, for HDVs, pedestrians constitute 15% and other vehicles 63% of accident involvements, as depicted in Figures 6 (a,b,c,d), respectively.

Accident outcomes:

Line 255-257:

When examining the outcomes of accidents, AVs slightly more often lead to either no injuries or minor injuries compared to HDVs, with the former occurring in 94% of AV accidents versus 92% in HDV accidents.

Comment 20:

Line 217-221, what are the factors that are included in the model but not significant?

Response:

We appreciate your question regarding the factors not significant in the model. In the manuscript, Table 2 enumerates all the variables that were found to be significant and thus included in the final model. Regarding the nonsignificant factors, these were identified during the model process but did not meet the statistical threshold

for inclusion in the final model. These nonsignificant variables are not listed in Table 2 but were considered during the initial stages of descriptive analysis listed in Table 1. These not significant variables are shown as follows:

Road and environment	
Weather	Clear (1 if weather is clear, 0 otherwise)
	Cloudy (1 if weather is cloudy, 0 otherwise)
	Fog (1 if weather is fog, 0 otherwise)
Road Condition	Normal road conditions (1 if there are no unusual conditions like construction, maintenance, or obstruction, 0 otherwise)
	Traffic event/work zone (1 if it is traffic event/work zone, 0 otherwise)
	Reduced Roadway Width (1 if road condition is reduced roadway width, 1 otherwise)
Road Surface	Dry surface (1 if surface is dry, 0 otherwise)
	Wet surface (1 if surface is wet, 0 otherwise)
	Snow/slush/ice (1 if surface is covered with snow/slush/ice, 0 otherwise)
Lighting	Daylight (1 if daylight, 0 otherwise)
	Dark– lighting (1 if it is dark– lit, 0 otherwise)
	Dark– not lighting (1 if it is dark– not lit), 0 otherwise)
Pre-accident conditions	
Associated Factor	Inattention (1 if driver is distracted, 0 otherwise)
	Entering/Leaving Ramp (1 if driver is Entering/Leaving Ramp, 0 otherwise)
	Defective Vehicle Equipment (1 if Vehicle Equipment is defective, 0 otherwise)
	Uninvolved Vehicle (1 if Vehicle is uninvolved, 0 otherwise)
	No dangerous driver behavior (1 if there is no dangerous driver behavior, 0 otherwise)
Pre-accident movement	Stopped (1 if pre-accident movement is stopped, 0 otherwise)
	Slowing (1 if pre-accident movement is Slowing, 0 otherwise)
	Changing lanes (1 if pre-accident movement is changing lanes, 0 otherwise)
	Into opposing lane (1 if pre-accident movement is into opposing lane, 0 otherwise)
	Parked (1 if pre-accident movement is parked, 0 otherwise)
	Travelling wrong way (1 if pre-accident movement is travelling wrong way, 0 otherwise)

	Pedestrian or bicycle (1 if accident with pedestrian or bicycle, 0 otherwise)
Accident Participation	Other participants (1 if accident with other participants, 0 otherwise)
	Vehicle (1 if accident with another vehicle, 0 otherwise)
Accident type	
	Head-on (1 if accident type is Head-on, 0 otherwise)
	Sideswipe (1 if accident type is Sideswipe, 0 otherwise)
	Hit-object (1 if accident type is Hit-object, 0 otherwise)
Accident outcomes	
Severity	No-injury (1 if injury severity outcome is no-injury, 0 otherwise)
	Minor (1 if injury severity outcome is minor, 0 otherwise)

Comment 21:

Line 221: 'This indicates a higher probability of an AV accident occurrence when these variables are present':

Comment: what do 'these variables are present' means, what is the direction? Do you mean that during dawn/dusk, and when turn indicator are present on the other vehicle, AV accident occurrence are higher? This is rather strange, do you know whether the turn indicator provided by the other vehicles are valid?

Response:

'these variables are present' means the presence of the dawn/dusk and turning variables affects the likelihood of AV accidents. This observation is based on the calculated coefficients and 95% confidence level, which suggests that AVs have a higher likelihood of accidents involvement under these specific conditions during dawn/dusk conditions or when the AV is turning. The turning variable includes both left, right, and U turns. The expression is revised in the manuscript:

Line 271-274:

This indicates a higher Odd of an AV accident occurrence when these variables are significant in the random parameter logit model. Furthermore, there are several variables that demonstrate high significance and exhibit negative coefficients, suggesting a reduced likelihood of an AV accident when these factors are significant.

Comment 22:

Line 224-226: I realized that the authors use 'indicator' as 'factor' perhaps they are redundant because they all are factors. Perhaps the comment above 'turn indicator' refers to a turning situation of AV (or other vehicle)

rather than the turn signals? Anyway, There are a lot of terms here that I do not understand (i.e., moderate severity, proceeding straight indicator, run-off road, backing indicator)

Response:

We have revised the language for greater clarity. The term 'indicator' has been replaced with 'condition' throughout the manuscript to avoid any confusion. Regarding the term 'turn indicator', it indeed refers to a situation where an AV (or another vehicle) is in the process of turning, and not to the vehicle's turn signals. To further clarify the terms that were identified as unclear, below a supplementary table that provides definitions and explanations for these, and other terms used in our study. It was not added to the manuscript due to space limitation.

Description and explanation for the variables in this paper

Variable type	Variable Value	Variable Description
Road and environment		
Road Condition	Normal road conditions (1 if there are no unusual conditions like construction, maintenance, or obstruction, 0 otherwise)	Self-explanatory.
	Traffic event/work zone (1 if there are traffic event/work zone, 0 otherwise)	If a prior traffic event such as traffic accidents, disabled vehicles, and spilled cargo that adversely affected normal traffic operations separate from the reported incident or work zone.
	Temporary reduced Roadway Width (1 if road condition is reduced roadway width, 1 otherwise)	For example, snow drifts, dirt slides, construction zones, etc.
Pre-accident conditions		
Associated Factor	Inattention (1 if driver is distracted, 0 otherwise)	It was an associated factor in the collision. For example, adjusting radio, lighting a cigarette, conversing with passengers, etc.
	Entering/Leaving Ramp (1 if driver is Entering/Leaving Ramp, 0 otherwise)	It includes collisions occurring on the ramp. For example, a driver starts to enter an on-ramp at an intersection and collides with another vehicle.
	Defective Vehicle Equipment (1 if Vehicle Equipment is defective, 0 otherwise)	For example, brakes, headlights, tread depth, etc.
	Uninvolved Vehicle (1 if Vehicle is uninvolved, 0 otherwise)	The involved party which claims that another vehicle contributed to the collision.
	No dangerous driver behavior (1 if there is no dangerous driver behavior, 0 otherwise)	Self-explanatory.
Pre-accident movement	Stopped (1 if pre-accident movement is stopped, 0 otherwise)	A vehicle not moving but, on the roadway, (excludes shoulder). A stalled, disabled, or abandoned vehicle on a roadway is considered stopped.
	Proceeding straight (1 if pre-accident movement is proceeding straight, 0 otherwise)	A vehicle proceeding straight ahead. A vehicle following the curve of a roadway is coded proceeding straight.
	Run-off Road (1 if pre-accident movement is Run-off Road, 0 otherwise)	If the motor vehicle left the road (includes paved or unpaved shoulder) prior to the collision and before evasive action began. This includes vehicles which would have left the road had their movement not been halted by colliding with protective barriers such as guardrails, concrete walls, or median barriers which are generally placed adjacent to, but outside the road.
	Turn (1 if pre-accident movement is turning left or right, 0 otherwise)	A vehicle making a turn at an intersection or into a private drive, or a vehicle stopped within an intersection preparing to turn. This includes an illegal turning movement, such as a turn when prohibited or when out of position. This excludes any lane change or turning movement to pass other vehicles.
	Backing (1 if pre-accident movement is backing, 0 otherwise)	A vehicle backing up, except when associated with parking.

	Slowing (1 if pre-accident movement is Slowing, 0 otherwise)	A vehicle in the process of slowing or stopping its forward movement. Speed is not a factor in determining whether this movement applies.
	Changing lanes (1 if pre-accident movement is changing lanes, 0 otherwise)	A vehicle making a lane change on a roadway divided into two or more clearly marked lanes for traffic in one direction.
	Entering traffic lane (1 if pre-accident movement is entering traffic lane, 0 otherwise)	A vehicle entering a designated lane of traffic from a shoulder, median, parking strip, alley, or private drive. Usually this is a forward movement, but it may include a backing movement associated with leaving a parked position.
	Into opposing lane (1 if pre-accident movement is into opposing lane, 0 otherwise)	A vehicle making an involuntary or unplanned movement into an opposing lane of traffic on a two-way road.
	Parked (1 if pre-accident movement is parked, 0 otherwise)	A vehicle not moving and outside of a traffic lane. This includes a vehicle stopped on a shoulder or in another area where parking is designated or permitted or a motor vehicle parked illegally but otherwise outside a traffic lane, such as blocking a driveway, beside a fire hydrant, or in a loading zone.
	Travelling wrong way (1 if pre-accident movement is travelling wrong way, 0 otherwise)	A vehicle proceeding in the opposite direction of traffic.
Accident Participation	Pedestrian (1 if accident with pedestrian, 0 otherwise)	A collision involves a bicycle or a motor vehicle in-transport and a pedestrian. Includes a person in or operating a coaster wagon, scooter, sled, skateboard, wheelchair, motorized wheelchair, or “electric personal assistive mobility device”.
	Other participants (1 if accident with other participants, 0 otherwise)	A collision involves a motor vehicle in-transport which comes in contact with another motor vehicle upon the same roadway or upon roadways within an intersection. Falling loads, detached trailers, etc., are considered part of the original motor vehicle.
	Vehicle (1 if accident with another vehicle, 0 otherwise)	A collision involves a motor vehicle in-transport which leaves the roadway and collides with another motor vehicle in-transport on another roadway. For example, a vehicle crosses a median strip and collides with a vehicle on an opposing roadway.
Accident type		
	Head-on (1 if accident type is Head-on, 0 otherwise)	Two motor vehicles, approaching from opposite directions, make direct contact. For example, the front of one vehicle collides with the front of another. Or prior to impact, one vehicle skids sideways, causing the side of the skidding vehicle to collide with the front of the other.
	Sideswipe (1 if accident type is Sideswipe, 0 otherwise)	One motor vehicle strikes the side of another with a glancing blow. For example, two vehicles are proceeding in the same direction or from opposite directions, and the side of one vehicle strikes the side of the other.
	Rear-end (1 if accident type is Rear-end, 0 otherwise)	Two motor vehicles, traveling in the same direction, make direct contact. For example, the front of one vehicle strikes the rear of another vehicle, or Vehicle #1 approaches Vehicle #2 from the rear and skids sideways during a braking action, causing the side of Vehicle #1 to strike the rear of Vehicle #2.
	Broadside (1 if the front of one vehicle slams into the side of another vehicle, 0 otherwise)	One motor vehicle strikes another vehicle at an angle greater than that of a sideswipe.
	Hit-object (1 if accident type is Hit-object, 0 otherwise)	A motor vehicle strikes a fixed object or other object.

Accident outcomes

Severity

No-injury (1 if injury severity outcome is no-injury, 0 otherwise)	Self-explanatory.
Minor (1 if injury severity outcome is minor, 0 otherwise)	This contains other visible injuries and complaint of pain. An injury, other than a fatal or severe injury, which is evident to observers at the scene of the collision. Other visible injuries include: (1) Bruises, discoloration, or swelling. (2) Minor lacerations or abrasions. (3) Minor burns. This classification could also contain authentic internal, other non-visible injuries, and fraudulent claims of injury. "Complaint of Pain" includes: (1) Persons who seem dazed, confused, or incoherent (unless such behavior can be attributed to intoxication, extreme age, illness, or mental infirmities). (2) Persons who are limping, or complaining of pain or nausea, but do not have visible injuries. (3) Any person who may have been unconscious, as a result of the collision, although it appears he/she has recovered. (4) Persons who say they want to be listed as injured but do not appear to be so.
Moderate (1 if injury severity outcome is moderate, 0 otherwise)	An injury, other than a fatal injury, that includes the following: (1) Broken or fractured bones. (2) Dislocated or distorted limbs. (3) Severe lacerations. (4) Skull, spinal, chest or abdominal injuries that go beyond "Other Visible Injuries." (5) Unconsciousness at or when taken from the collision scene. (6) Severe burns.
Fatal (1 if injury severity outcome is fatal, 0 otherwise)	Death as a result of injuries sustained in a collision, or an injury resulting in death within 30 days of the collision.

1 Comment 23:

2 Figure 3 to 6, it will be useful to provide (N = and % information to each category)

3 **Response:**

4 We have updated these figures to include both the number of observations (N =) and the percentage (%)
5 information for each category, enhancing the clarity and informational value of the data presented.

(a) HDV accidents (35,133)

(b) ADS+ADAS accidents (2100)

(c) ADAS accidents (1001)

(d) ADS accidents (1099)

Figure 6. Distribution of the factors influencing accidents of various vehicle types.

Comment 24:

Line 249-250: Perhaps it is due to the help of advanced algorithms, but do the authors know what is the chances of HDV hitting HDV? If HDV hitting AV is significantly higher than HDV hitting HDV, it could also be interpreted as AV was not designed in a way that can be understood and predicted by the human driver following behind (I.e., abrupt stopping with no valid reason and in weird places?)

Response:

The chances of HDV hit other vehicles and other vehicles hit HDV are added to the manuscript:

Line 286-291:

Our data indicates that for HDVs, the rear-end stands at 45.33% (other vehicles hit HDV), while head-on accidents (HDV hit other vehicles) occur at a rate of 32.68%. In contrast, AVs have a slightly lower rear-end accidents (AV hit by other vehicles) rate of 39.19%, but a similar head-on accidents (other vehicles hit by AV) rate of 32.71%. This suggests that while AVs have a marginally reduced risk of being rear-ended compared to HDVs, their percentage to head-on accidents is almost the same.

Note: The crash report provides the damage area and pre-accident movement of the Subject Vehicles and Crash Partner. Thus, we can identify whether it is rear-end accident, and which vehicle hits the other. Inferring which vehicle hit the other in a head-on accident.

Comment 25:

Line 252. I do not follow: 'when in conventional mode, more than half of the AV vehicles are moving, and the others are stopping'.

Response:

When a HDV hit an AV, we see that for the accidents where the AV was moving (middle section), 62 cases (57%) involved AVs in autonomous mode while 46 cases (43%) involved AVs in conventional mode (left section). We also provide a clearer explanation in the manuscript:

Line 306-308:

For example, In Figure 7 (a), which details accidents where a HDV hit an AV, we see that for the accidents where the AV was moving (middle section), 62 cases (57%) involved AVs in autonomous mode while 46 cases (43%) involved AVs in conventional mode (left section).

Comment 26:

Line 257-258: in terms of accident severity, 82% of accidents occur as minor when HDV hit AV (does that mean 18% is severe?). This percentage is 67% when the AV hit HDV (does that mean 67% as minor but 33% as severe?). It seems to contradict the above that says HDV accidents are more severe?

Response:

To clarify,

18% is moderate and major injury severity when HDV hit AV for rear-end accidents.

33% is moderate and major injury severity when AV hit HDV for rear-end accidents.

It's important to note that these percentages specifically refer to rear-end accidents.

To address the perceived contradiction regarding the severity of HDV and AV accidents, from the Matched Case-Control Logistic Regression Model in Table 2. This is further supported by our odds ratio (OR) analysis.

Moderate Injury (OR = 0.609 for AV compared with HDV): This OR ratio suggests that the odds of a moderate injury occurring in an accident involving an AV is 0.609 times the odds of a moderate injury in an accident involving an HDV. Specifically, it implies that the risk of moderate injury is about 39% lower in AV accidents than in HDV accidents.

Fatal Injury (OR = 0.592 for AV compared with HDV): Similarly, this OR ratio indicates that the odds of a fatal injury in an AV accident are 0.592 times the odds in an HDV accident. The risk of fatal injury in AV accidents is approximately 41% lower than in HDV accidents.

Comment 27:

Figure 7: please include % in the figure. Also, it is perhaps important to clarify why these AVs were operating in conventional mode? Does it indicate that those are the more complex situations where AV cannot handle? And thus, the higher rate of accident in AV hit HDV driven in conventional mode could have been a bit of an unfair comparison? Worth discussing.

Response:

The percentage is added to the Figure 7.

Regarding your concern about why these AVs were operating in conventional mode, it is indeed an important point to consider. The accident report does not provide clear information about the reasons for disengagement. While we cannot definitively conclude whether AVs were operating in more complex situations based on our data, according to Dixit et al. in 2016, 56.1% of disengagements were attributed to system failures, 26.57% were initiated by the driver, and 9.89% were related to road infrastructure issues¹⁴. The precise reasons behind the higher rate involving AVs hitting HDVs in conventional mode cannot be definitively determined. Our analysis focused on comparing the distribution differences between conventional and autonomous modes for rear-end accidents. To gain a more in-depth understanding, further analysis can be conducted once more detailed reporting becomes available.

Comment 28:

Please clarify that the authors first make sure that the accidents happened in the same location (or within 5 miles radius), same day of the week and time of day, before matching with the other variables such as those mentioned in the table 2? Were traffic patterns assumed based on the same day of the week and time of day?

Response:

In response to your concern, we would like to clarify that the initial step in our methodology involved ensuring that the accidents occurred either at the same location or within a 5-mile radius. Additionally, we meticulously matched these cases with controls based on the same day of the week and time of day to maintain consistency in environmental and temporal factors. This approach was adopted with the underlying assumption that traffic

patterns remain relatively consistent for the same time periods and days of the week. This assumption is foundational to our analysis and has been explicitly stated in the revised manuscript as follows:

Line 325-326:

We assumed that cases and controls were under similar traffic patterns based on the controlled time and space.

Comment 29:

Line 289: 'This indicates a lower likelihood of an AV accident in rainy weather compared to an HDV accident'. I appreciate that the authors mentioned that these vehicles are seldom tested in adverse weather. Therefore, it pretty much goes back to the previous question on the 'accident vs no accident ratio' under each condition for these vehicle type. Understanding the discrepancy would provide a better understanding on the likelihood.

Response:

Direct comparison between AV and HDV crashes is still not viable as the difference in exposure and number of vehicles of both types is extremely unbalanced. To compare the likelihood of AVs and HDVs, we incorporated Annual Average Daily Traffic (AADT) data under varying weather conditions from the California Traffic Census Program. This enabled us to calculate the accident rate under different traffic conditions. The method for calculating accident rates takes into consideration vehicle exposure by including AADT. The accident rates at segments can be determined using the equation below:

$$R_{section} = \frac{A * 1,000,000}{365 * T * V_{section} * L}$$

Where

$R_{section}$ is the accident rate for the section.

A is the number of reported accidents during the time period.

T is the number of years considered.

$V_{section}$ is AADT or annual average daily traffic, vehicles per day.

L is the length of section, miles.

For intersections, the accident rates can be determined using the equation below:

$$R_{spot} = \frac{A * 1,000,000}{365 * T * V_{spot}}$$

Where

R_{spot} is the accident rate for the spot.

A is the number of reported accidents during the time period.

T is the number of years considered.

V_{spot} is AADT or annual average daily traffic ($V_{spot} = (AADT_{minor\ roads} + AADT_{major\ roads})$), vehicles per day.

The accident rate by varying weather is shown as in the chart below. Accident rates are classified into four categories: Clear, Cloudy, Rain and Fog. HDVs show a higher incidence of accidents under clear, rain and fog conditions. Under cloudy conditions, HDVs and AVs exhibit similar accident rates.

This accident rate is consistent with the result of matched case-control that the odds of an AV (ADS) accident occurring in rainy weather are 0.336 times. This indicates a lower likelihood of an AV accident in rainy weather compared to an HDV accident.

Comment 30:

The line in 300 to 302 quickly contradicts with 297-298. Authors should be very careful in making these claims on sensor capabilities, although I understand citations were provided. But how accurate these really are? Not only in terms of detection but intention recognition? especially followed by the discussion on Situation Awareness below.

Response:

In response to your concerns regarding the accuracy of the sensor capabilities in AVs, we have revised the text to provide a clearer and more nuanced explanation about the sensor capabilities. The revised section now reads:

Line 352-359:

AVs have the capability to react in under 500 milliseconds. Compared with that, the average human reaction time falls behind the slowest steering reaction of an AV by 12% and is 2.6 times slower compared to the slowest emergency braking response of such vehicles³². Although adverse weather can increase the likelihood of

potential failures or loss of AV sensors, recent innovations in visual algorithms ³³⁻³⁵, coupled with the combined use of cameras, LIDAR ³⁶, GNSS³⁷, and radar sensors ³⁸, are crafted to recognize pedestrians and vehicles under varying weather scenarios, such as cloudiness, snow, rain, and darkness ^{39,40}. This offers solutions to the challenges associated with operating in less-than-ideal conditions.

Comment 31:

Line 316 to 327: interesting, perhaps expand what situation awareness is and provide a citation, but this explanation is rather vague, do the authors know why is turning somehow 'more difficult' than other situation?

Response:

We appreciate your suggestion for elaboration on situation awareness and its challenges in AVs. Accordingly, we have enhanced our manuscript with a more detailed explanation and relevant citations.

Situation awareness in AVs is now more comprehensively described as follows (Referenced as citation 45 in the manuscript):.

Line 385-387:

Situational awareness of AVs can be defined as the ability of these vehicles to perceive essential elements in their surroundings, understand the importance of these elements, and anticipate their future state or changes⁴⁸.

Regarding the specific query about the complexity of turning in AVs, we have provided a detailed explanation:

Line 387-391:

The complexity of turning in autonomous driving scenarios arises from three primary challenges: choosing the appropriate lane (target lane selection), devising and computing a safe and efficient path (trajectory planning and calculation), and executing the turn while adjusting to dynamic conditions (vehicle controlling and tracking)

⁴⁹.

Reviewer #2

General Comment:

Summary of paper content

The work is analyzing accident data of autonomous vehicles statistically and compares them with accident data of human driven vehicles. It uses a matched case-control analysis in which it uses the AV data as cases and the HDV data as controls. It tries to derive in which situations AV accidents are more (less) likely to happen than HDV accidents.

Novelty & Relevance

The topic of analyzing accidents of autonomous vehicles is highly relevant to understand the potentials of autonomous driving in comparison with human driving. Therefore, the paper discusses a very interesting question that isn't yet analyzed sufficiently well in previous work. The results provide some novel scientific insights.

Organization & Readability

Comment 1:

line 47: please introduce the full name of the NHTSA before using the abbreviation. I guess you meant the National Highway Traffic Safety Administration but that might not be clear to all readers.

Response:

As suggested, we have introduced the full name of the National Highway Traffic Safety Administration (NHTSA) before employing its abbreviation to ensure that all readers are aware of its meaning.

Comment 2:

Line 171: please add an 's' at the end of the word "depend"

Response:

The suggested correction has been implemented, where an 's' has been added to the word "depend."

Comment 3:

line 174: I am not sure whether "gauge" is the perfect word here. You might like to replace it by another expression like "estimate" or "derive"

Response:

The term "gauge" has been replaced with "estimate" to better convey the intended meaning.

Comment 4:

Line 243: what does the word "ed" mean?

Response:

We apologize for the confusion caused by typos. The term "ed" was inadvertently included in the text. It was intended to be part of the word "revealed," thus contributing to the meaning of the sentence. This error is corrected in the revised manuscript.

Technical Soundness

Comment 1:

Section 3.1: It would be interesting to get more information on the data. Which scenarios do we find in the data, highways, rural roads, urban driving? Which collision types do they contain to which extend?

Response:

To address your query, we have expanded our analysis to include a detailed description of accident types across different driving scenarios, such as highways, intersections, and rural roads.

Lines 171-183:

The distributions of accident types (head-on, sideswipe, rear-end, broadside) for vehicle categories (HDV and ADS) are shown in Figure 3, which visually illustrates the frequency and proportion of each accident type in the respective locations by vehicle categories.

Figure 3. Distribution of the accident's types and scenarios

For HDV accidents, intersections are the primary locations (significantly higher than other HDV accident location types with $F= 5.1043$ and $p= 0.0166$), where 48.7% of HDV accidents at intersections are rear-end, making it the most common type of accidents. Urban streets are the second most common scenario, with head-on

accounting for 48.0% of HDV street accidents. Conversely, ADS accidents occur more frequently on urban streets (significantly higher than other ADS accident location types with $F= 10.4982$ and $p= 0.0011$), where 47.4% of ADS street accidents were head-on. Accidents at intersections are the second most common for ADS, with rear-end making up 42.8% of these ADS intersection accidents.

Comment 2:

Section 3.2, Table 1: I would have expected some more criteria being part of the list, especially (i) vehicle speed, and (ii) right-of-way at intersections (i.e. yield/stop/priority/traffic lights). Why didn't you consider these?

Response:

We would like to clarify that our dataset exclusively includes pre-accident movement data for ADS and ADAS, such as Stopped, Proceeding, Run-off and Travelling wrong way. Unfortunately, it does not include data related to right-of-way at intersections, such as yield signs, stop signs, priority signals, or traffic lights. This part is added as limitation in the future work:

Line 459-461:

It would also be crucial in the future to incorporate data about right-of-way at intersections, encompassing yield signs, stop signs, priority signals, and traffic lights, to enhance the comparative analysis between AV and HDV.

For the pre-accident vehicle speed, we have added a heatmap in the manuscript (Figure 5), visually representing the average pre-accident speeds of these vehicles. This heatmap provides a comprehensive view of the speed patterns across different days of the week and times of the day, comparing ADS and ADAS vehicles.

(a) ADAS Average Pre-accident Speed

(b) ADS Average Pre-accident Speed

Figure 5. Distribution of the Pre-accident Speed

Line 227-232:

To enhance the understanding of pre-accident speeds, heatmaps that visually represent these speed patterns for ADS and ADAS vehicles is shown in Figure 5. This chart offers a detailed comparison of how speeds vary across different days of the week and at various times of the day. This trend can be attributed to the fact that ADAS is primarily designed for highway use, leading to a higher average pre-accident speed when compared to ADS vehicles, which are designed for a complex urban driving scenario.

Comment 3:

Section 4: The description of your matched case-control logistic regression model was very unclear to me. I could not fully follow the description so I will go through the text and tell you what I understood and what looked strange to me.

Response:

Based on your comments, we have thoroughly reviewed and revised this section.

Line 128-151:

Conditional logistic regression is a variant of logistic regression that specifically tackles the issue of stratification within matched case-control studies²⁵. In this research, there are N strata denoted by $i = 1, 2, \dots, N$. Each stratum has one AV accident case sample and k HDV accident control samples denoted by $j = 1, 2, \dots, k$. The conditional likelihood for the i th strata depends on the probability of the total number of cases (AV accident case samples) and controls (k HDV accident control samples) recorded in the stratum²⁶. $P_{ri}(x_{ji})$ refers to the probability of the j th samples in the i th stratum is a AV accident where $x_{ij} = (x_{1ij}, x_{2ij}, \dots, x_{pij})$ is determined by a vector of variables (x_1, x_2, \dots, x_p) . A logistic regression model with linear parameters is employed to estimate the likelihood of an accident, as described by Abdel-Aty et al. (2004)²⁷ :

$$\text{logit}|P_{ri}(x_{ji}) = (a_i, b_1x_{1ji}, \dots, b_kx_{pji}) \quad (1)$$

The controlled variables used to create strata are reflected in the intercept term. To incorporate the impact of stratification in the analysis, it is possible to construct a conditional log-likelihood. This log-likelihood function comprises multiple terms, each representing the conditional probability of an accident occurring within a specific stratum²⁸. The following equation presents the formula for the conditional likelihood function, as stated by Abdel-Aty et al. (2004)²⁷ :

$$E(\beta) = \prod_{i=1}^N [(1 + \sum_{j=1}^k \exp\{\sum_{u=1}^p \beta_u(x_{uji} - x_{uoi})\})]^{-1} \quad (2)$$

The coefficients' estimates in Equation (1) are identical to the maximum likelihood function values in Equation (2). These estimates are log-odds ratios that can provide an approximation of the relative risk of an accident and are also referred to as hazard ratios (i.e., the ratio of odds for accident occurrence versus non-occurrence). The hazard ratio is calculated by raising the exponential value to the coefficient's power. For a dummy variable, the odds ratio is a statistic defined as the ratio of the odds of the case. The odds ratio can be written as

$$OR(x_k) = \frac{\Pr(y_{i0}=1, x_k=1, Z) / [1 - \Pr(y_{i0}=1, x_k=1, Z)]}{\Pr(y_{i0}=1, x_k=0, Z) / [1 - \Pr(y_{i0}=1, x_k=0, Z)]} = \exp(\beta_k) \quad (3)$$

where, Z represents the vector of explanatory variables excluding x_k . β_k is the estimated coefficient for x_k .

Comment 4:

Lines 159-165: seems to be clear to me. However, I am wondering whether the application of using AV accidents as cases and HDV accidents as controls really is appropriate. In references 23 and 24 accidents are used as cases and non-accidents as controls and the idea is to find out the most relevant factors to explain the occurrence of accidents. Here, you deal with accident cases only and compare AV accidents with HDV accidents. This will not allow you to find out which factors are relevant for the occurrence of accidents in general but only allows you to compare the distributions of AV accidents and HDV accidents over various situations. I doubt that you can draw the conclusions that you claim in Section 5 since you completely ignore the non-accident cases and so you ignore the prior probability of accidents of AVs and HDVs (i.e. the number of accidents per distance traveled).

Response:

Our aim is to explore the differential characteristics of accidents involving AVs vs HDVs, rather than comparing accidents and non-accidents. So, the matched case-control study can be used to examine the impact of exogenous variables on accident risk for different vehicle types (AV (ADS) and HDV). The distribution of AV accidents over various situations differs from the distribution of HDV accidents can also be concluded from the matched case-control study. Please note that there is an unlimited number of non-accident cases, and they don't have any of the variables of interest to conduct matched control design. Comparison of accident rates as shown below could be sufficient in that case.

Additionally, the prior probability of accidents (i.e., the frequency of accidents per distance traveled) for AVs and HDVs is also crucial to this study. To delve deeper into the accident likelihood under non-accident conditions for AVs, we sourced data on autonomous vehicle miles traveled (AVMT), disengagement accidents, and accident occurrences from the publicly available California DMV AV accident and disengagement reports. We calculated the disengagement rate per 1,000 miles and the accident rate per 1,000 miles.

Number of AV AVMT, disengagement and Accidents

Year	AVMT	Disengagement	Disengagement/1000 mi	Accident	Accident /1000 mi
2017	483786	11281	23.32	22	0.05
2018	1971474	66049	33.5	44	0.02
2019	2636198	1972	0.75	67	0.03
2020	1565353	335	0.21	20	0.01
2021	4051850	8216	0.2	117	0.03
2022	5964804	2376	0.04	150	0.03

Number of AV AVMT and Accidents

For the HDV group, we gathered data on Motor Vehicle Miles of Travel (MVMT) from the SWITRS Annual Report of Fatal and Injury Motor Vehicle Traffic Collisions. This information allowed us to compute the Accident rate per 1 million miles for HDVs.

Number of HDV MVMT and Accidents

Year	MVMT	Accident	Accident /1 million mi
2010	32777000000	161094	0.05
2011	32503200000	159115	0.05
2012	32654700000	159696	0.05
2013	32917400000	156909	0.05
2014	33466400000	162742	0.05
2015	33984300000	178669	0.05
2016	34285300000	195347	0.06
2017	34430400000	193564	0.06
2018	34719400000	191971	0.06
2019	35115100000	187211	0.05

*Data after 2019 is Not yet available

The method for calculating accident rates takes into consideration vehicle exposure by including AADT. The accident rates at segments can be determined using the equation below:

$$R_{section} = \frac{A * 1,000,000}{365 * T * V_{section} * L}$$

Where

$R_{section}$ is the accident rate for the section.

A is the number of reported accidents during the time period.

T is the number of years considered.

$V_{section}$ is AADT or annual average daily traffic, vehicles per day.

L is the length of section, miles.

For intersections, we use the total entering AADT as the variable V. The accident rates at intersections can be determined using the equation below:

$$R_{spot} = \frac{A * 1,000,000}{365 * T * V_{spot}}$$

Where

R_{spot} is the accident rate for the spot.

A is the number of reported accidents during the time period.

T is the number of years considered.

V_{spot} is AADT or annual average daily traffic ($V_{spot} = (AADT_{minor\ roads} + AADT_{major\ roads})$), vehicles per day.

The accident rate and number of accident comparison by road type is shown as the chart below. Accident rates are classified into four categories: Highways, Intersections, Rural Roads, and Streets. HDVs show a higher incidence of accidents on highways, intersections, and streets, particularly on highways. For rural roads, HDVs and AVs exhibit similar accident rates. Across all road types, HDVs consistently record significantly higher accident figures than AVs.

This distribution under different scenarios such as weather, is consistent with the results of matched case-control study conducted in Section 5. The accident rate is consistent with the result of matched case-control that the odds of an AV (ADS) accident occurring in rainy weather are 0.336 times, as the chart shown below. This indicates a lower likelihood of an AV accident in rainy weather compared to an HDV accident.

Direct comparison between AV and HDV crashes is still not viable as the difference in exposure and number of vehicles of both types are extremely unbalanced.

Comment 5:

Line 169: I looked in reference 25 but I did not understand why it is cited here.

Response:

The reference is the first to utilize the matched case control design used frequently in medicine and epidemiology to traffic safety. It provides a good description of the methodology and explains how it could be applied to traffic safety. We appreciate your observation and acknowledge the need for clarity. The reference has been carefully reviewed and updated.

Comment 6:

Line 170: Are “small k” and “capital K” the same variable or different variables? I assume it should be the same.

Response:

To clarify, they indeed represent the same variable. We have now revised the manuscript to ensure consistency in the use of “small k” throughout the paper to avoid any ambiguity.

Comment 7:

Line 170: You mention that one stratum contains k HDV accident samples. How do you use the word "sample" here? I would interpret a sample as a set of accidents which would mean that one stratum consists of one AV accident and k sets of HDV accidents. Or did you mean "sample element" rather than "sample" (i.e. a single HDV accident)? Then, it would make more sense. Please check your wording.

Response:

In a case-control study, the term "sample" means all the cases or controls within a particular stratum, not the total of all strata combined.

To improve clarity, it has been clarified in the paper:

Line 187-192:

Samples generally refer to the groups of accidents selected for comparison within each stratum. Case samples are specific accidents who have the outcome that is the focus of the study. Control samples are accidents who do not have the specific outcome being studied. In this study, ‘AV accident case sample’ consists of AV accidents within each stratum, and the ‘HDV accident control samples’ consist of the HDV accidents within the same stratum.

Comment 8:

Line 171-172: You use the word “observation” and the expression “the number of accidents recorded in this stratum”. If I assume the second interpretation of line 170 (see above), the number of accidents recorded in this

stratum is $k+1$. So, what is the number of observations? I would guess, it is the same, i.e. $k+1$. But why do you use two different expressions for the same? That remains unclear to me.

Response:

To clarify, the term "observations" is indeed synonymous with "samples" as previously mentioned in the text. The revised expression is given as follows:

Line 129- 134:

There are N strata denoted by $i = 1, 2, \dots, N$. Each stratum has one AV accident case samples and k HDV accident control samples denoted by $j = 1, 2, \dots, k$. The conditional likelihood for the i th stratum depends on the probability of the total number of cases (AV accident case samples) and controls (k HDV accident control samples) recorded in the stratum²⁴. $P_{ri}(X_{ji})$ refers to the probability of the j th samples in the i th stratum is a AV accident where $X_{ij} = (X_{1ij}, X_{2ij}, \dots, X_{pij})$ is determined by a vector of variables (X_1, X_2, \dots, X_p) .

Comment 9:

Line 172-175: You introduce $P_{ri}(X_{ji})$ as the probability that the j -th observation in the i -th stratum is an accident. According to what was said before all observations are accidents. So $P_{ri}(X_{ji})=1$. Or are there some non-accidents in the data set? Or should $P_{ri}(X_{ji})$ be the probability that the j -th observation in the i -th stratum is an AV accident? Please clarify it.

Response:

The definition of $P_{ri}(X_{ji})$ is given as follows:

Line 133- 134:

$P_{ri}(X_{ji})$ refers to the probability of the j th samples in the i th stratum is a AV accident where $X_{ij} = (X_{1ij}, X_{2ij}, \dots, X_{pij})$ is determined by a vector of variables (X_1, X_2, \dots, X_p) .

Line 129- 131:

There are N strata denoted by $i = 1, 2, \dots, N$. Each stratum has one AV accident case samples and k HDV accident control samples denoted by $j = 1, 2, \dots, k$.

Comment 10:

Line 172-175: First, you are using X_{ji} , later on you are using X_{ij} . Why do you exchange the indexes?

Response:

We have revised the notation to ensure consistency throughout. The corrected notation now consistently uses as follows:

Line 137:

$$\text{logit}|P_{ri}(x_{ji}) = (a_i, b_1x_{1ji}, \dots \dots b_kx_{pji})$$

Comment 11:

Line 182: are "small x_{uji} " in (2) the same as "capital X_{uji} " in (1) or not?

Response:

To clarify, the "small x_{uji} " mentioned in equation (2) and the "capital X_{uji} " in equation (1) are the same. We have revised the manuscript to ensure consistent and clear notation, thereby eliminating any potential confusion.

Comment 12:

By the way, the whole mathematical notation in this chapter is rather sloppy and confusing. A strict distinction between probabilities, frequencies, random events, random variables, estimates, and other terms would be rather helpful to improve readability. In its present form I don't feel that the notation is sufficient for publication. E.g. the term $P_{ri}(X_{ji})$ would indicate that X_{ji} is a random event since probabilities are only defined for random events. It might also be a random variable (the choice of a capital letter would indicate this) in a sloppy mathematical notation (which, however, you find rather often in literature), however, it is hard to assess how the term would look like in strict mathematical notation. The term $X_{ij}=(X_{1ij}, \dots)$ however would indicate that X_{ij} (or X_{ji} ?) is just a data point and neither a random variable nor a random event.

Response:

To address your concerns, we undertook a thorough revision of the notation throughout the section. This includes clearly distinguishing between probabilities, frequencies, random events, random variables, and estimates.

Specifically, we reevaluated the use of terms like $P_{ri}(X_{ji})$ to ensure that they accurately represent the intended mathematical concepts, be it a random event or a random variable.

Line 128-152:

Conditional logistic regression is a variant of logistic regression that specifically tackles the issue of stratification within matched case-control studies²⁵. In this research, there are N strata denoted by $i = 1, 2, \dots, N$. Each stratum has one AV accident case sample and k HDV accident control samples denoted by $j = 1, 2, \dots, k$. The conditional likelihood for the i th strata depends on the probability of the total number of cases (AV accident case samples) and controls (k HDV accident control samples) recorded in the stratum²⁶. $P_{ri}(x_{ji})$ refers to the probability of the j th samples in the i th stratum is a AV accident where $x_{ij} = (x_{1ij}, x_{2ij}, \dots, x_{pji})$ is determined by a vector of variables (x_1, x_2, \dots, x_p) . A logistic regression model with linear parameters is employed to estimate the likelihood of an accident, as described by Abdel-Aty et al. (2004)²⁷ :

$$\text{logit}|P_{ri}(x_{ji}) = (a_i, b_1x_{1ji}, \dots \dots b_kx_{pji}) \quad (1)$$

The controlled variables used to create strata are reflected in the intercept term. To incorporate the impact of stratification in the analysis, it is possible to construct a conditional log-likelihood. This log-likelihood function comprises multiple terms, each representing the conditional probability of an accident occurring within a specific stratum²⁸. The following equation presents the formula for the conditional likelihood function, as stated by Abdel-Aty et al. (2004)²⁷ :

$$L(\beta) = \prod_{i=1}^N [(1 + \sum_{j=1}^k \exp\{\sum_{u=1}^p \beta_u (x_{uji} - x_{u0i})\})]^{-1} \quad (2)$$

The coefficients' estimates in Equation (1) are identical to the maximum likelihood function values in Equation (2). These estimates are log-odds ratios that can provide an approximation of the relative risk of an accident and are also referred to as hazard ratios (i.e., the ratio of odds for accident occurrence versus non-occurrence). The hazard ratio is calculated by raising the exponential value to the coefficient's power. For a dummy variable, the odds ratio is a statistic defined as the ratio of the odds of the case. The odds ratio can be written as

$$OR(x_k) = \frac{\Pr(y_{i0}=1, x_k=1, Z) / [1 - \Pr(y_{i0}=1, x_k=1, Z)]}{\Pr(y_{i0}=1, x_k=0, Z) / [1 - \Pr(y_{i0}=1, x_k=0, Z)]} = \exp(\beta_k) \quad (3)$$

where, Z represents the vector of explanatory variables excluding x_k . β_k is the estimated coefficient for x_k .

Comment 13:

Section 5 (and 6): As mentioned above, I doubt that you can conclude from your study that AV accidents are more (or less) likely in some situations than HDV accidents since you completely ignore the prior probabilities of accidents. The only conclusions that you may draw are that the distribution of AV accidents over various situations differs from the distribution of HDV accidents. I think this point should be made clearer. Otherwise, the reader gets an erroneous impression.

Response:

The prior probabilities of accidents are described in response to comment 4. Our aim is to explore the differential characteristics of accidents involving AVs and HDVs, rather than comparing accidents and non-accidents. The distribution of AV accidents over various situations differs from the distribution of HDV accidents is concluded from the matched case-control study. Direct comparison between AV and HDV crashes are still not viable as the difference in exposure and number of vehicles of both types is extremely unbalanced. That's why we provided all available AV accidents and compared between them and a comparable sample of HDVs. We also compared ADS with ADAS. Extensive research over the last few decades has identified the characteristics and details of HDV. Here we focus on showing how AV compares to HDV and potentially infer why AV accidents happen. In the near future, more reported AV crash data would become available and enable us to conduct more detailed analysis. We addressed in the manuscript at the beginning of the matched case-control study:

Line 335-339:

Our aim is to explore the differential characteristics of accidents involving AVs and HDVs, rather than comparing accidents and non-accidents. To examine the impact of exogenous variables on accident risk for different vehicle types, we conducted a matched case-control logistic regression model for AV (ADS) and HDV accidents. The distribution of AV accidents over various situations differs from the distribution of HDV accidents is concluded from the matched case-control study.

Comment 14:

Section 5 Figure 7: I was unable to interpret this figure. Please explain it in more detail.

Response:

We appreciate the reviewer's comment regarding the interpretability of Figure 7 in Section 5. In response to this feedback, we have revised the figure to enhance its clarity and added a more detailed explanation to the text.

Figure 7. Rear-end accident conditions between AV and HDV

Lines 300-311:

Figure 7 describes two conditions related to AV rear-end accidents: a: a HDV has hit an AV from behind (252) and b: an AV has hit an HDV from behind (67). The left side of the diagram starts with two conditions of HDV hit AV or AV hit HDV. The middle section shows the movement of vehicle: Moving or Stopping. The right side categorizes the severity of the accidents: Minor, Moderate and Major. The numbers indicated in each section

of the diagram correspond to the total count of each specific category. The width of each link connecting the sections of the diagrams represents the proportion of scenarios that fall into the subsequent category. For example, In Figure 7 (a), which details accidents where a HDV hit an AV, we see that for the accidents where the AV was moving (middle section), 62 cases (57%) involved AVs in autonomous mode while 46 cases (43%) involved AVs in conventional mode (left section). Among the 108 accidents categorized under the 'moving condition', there were 2 cases (2%) that resulted in major injuries, 18 cases (16%) that led to moderate injuries, and the remaining 88 cases (81%) involved minor injuries.

Comment 15:

line 293-294: AVs cannot react within milliseconds, at least not within a few milliseconds. Sensing, computing, and acting require time. For all technical systems that I know the reaction time is a couple of 100 milliseconds.

Response:

To clarify, the response time of AVs is revised as follows:

Line 352-355:

AVs have the capability to react in under 500 milliseconds. Compared with that, the average human reaction time falls behind the slowest steering reaction of an AV by 12% and is 2.6 times slower compared to the slowest emergency braking response of such vehicles³⁴.

Comment 16:

line 295-298: Bad weather conditions affect the perception abilities of cameras a lot, of lidar systems to some extent. Only radar systems and GNSS do not suffer that much from bad weather, but they have other shortcomings.

Response:

We appreciate the reviewer's comment. In response, we have expanded our literature review to address the safety performance of AVs more thoroughly under challenging weather conditions.

Lines 355-359:

Although adverse weather can increase the likelihood of potential failures or loss of AV sensors, recent innovations in visual algorithms³⁵⁻³⁷, coupled with the combined use of cameras, LIDAR³⁸, GNSS³⁹, and radar sensors⁴⁰, are crafted to recognize pedestrians and vehicles under varying weather scenarios, such as cloudiness, snow, rain, and darkness^{41,42}. This offers solutions to the challenges associated with driving in less-than-ideal conditions.

Comment 17:

Line 308-314: The explanation that AVs avoid rear-end accidents because they react faster is possible but not certain. Another explanation might be that AVs keep larger distances than HDVs in dense traffic. Without knowing more details about these accidents we cannot decide which explanation fits.

Response:

We agree that the possibility of AVs maintaining larger distances than HDVs in dense traffic is a valid and important consideration. To strengthen our argument, we have incorporated this perspective into our discussion, supported by relevant literature:

Lines 375-380:

In addition, the kinematic method used by ACC system keeps track of and regulates the space between vehicles, alerting drivers if this space becomes smaller than the safe limit. By ensuring that vehicles keep a consistent speed and spacing between vehicles, thus effectively reduces the risk of rear-end accidents⁴⁷. In contrast, HDVs tend to display greater variability in both speed and acceleration, a factor that significantly contributes to a higher incidence of rear-end and side-scratch accidents⁴⁸.

Comment 18:

line 319-321: Limited sensor ranges and limited coverage of the environment by sensors might also be an explanation.

Response:

We appreciate your insightful suggestions regarding the coverage of sensors in our manuscript. Following your advice, we have expanded the discussion on this topic, much better name substantiating our explanation with relevant literature references.

Line 391-396:

AVs rely on sensors and algorithms to perceive their surroundings and make driving decisions⁵⁰. However, these systems may not detect all obstacles and hazards, particularly in complex and dynamic driving scenarios like turning at intersections⁵¹. It is a significant challenge to generate sufficient information and achieve comprehensive detection of the surrounding environment from a single independent source due to limited sensor ranges and limited coverage of the environment by sensors in AV⁵².

Comment 19:

Line 322-323: Different AVs follow different driving strategies. Some might follow predefined strategies but there are also vehicles which calculate their behavior as a reaction to the present situation so that they are able to deal with unseen situations.

Response:

In the revised text, we now acknowledge these different mechanisms. On the one hand, we mention that some AVs are programmed to adhere to predefined rules and scenarios. On the other hand, we also discuss AVs that dynamically calculate their behavior in response to real-time situations. The revision is shown as follows:

Line 396-406:

Additionally, some AVs are programmed to follow predefined rules and scenarios, which may not encompass every possible driving situation⁵³. The modifications of scenarios can present difficulty for AVs in perceiving and responding to them, thereby raising the risk of an accident⁵⁴. Some vehicles calculate their behavior as a reaction to the present situation so that they may be able to deal with unexpected situations. However, multiple oncoming HDVs and the complexity of such driving scenarios are a considerable challenge for AVs such as unprotected left turns at intersections⁵⁵. These situations are complicated by factors like limited priority and variation in trajectories⁵⁶. AVs tend to be overcautious (such as having a longer startup delay during the turning at intersections)⁵⁷, which can lead to rear-end or sideswipe accidents with HDVs⁵⁸. Furthermore, multi-interactions caused by mixed flows aggravate uncertainties in detection, such as the superposition of distance and angle measurement error⁵⁹.

Comment 20:

line 324-327: Human drivers might have driven several hundred thousand of kilometers themselves. AVs are tested on millions or even more kilometers. Who of both has more driving experience?

Response:

We've revised our manuscript to explain that HDV may perform better than AV when turning from the perspective of HDV.

Line 406-411:

Conversely, HDVs can adapt and modify their speed more seamlessly than AVs⁶⁰. While AVs face difficulties with executing lane changes or turning in heavy traffic and lack psychological insight⁶¹. In addition, HDVs can anticipate pedestrian movements based on behavior and position based on their driving experience, exercising caution, whereas AVs may struggle with recognizing pedestrians' intentions, potentially leading to emergency braking or accidents due to a lack of understanding of social cues and psychological reasoning⁶¹⁻⁶³.

For the driving experience, we sourced data on autonomous vehicle miles traveled (AVMT), disengagement accidents, and accident occurrences from the publicly available California DMV AV accident and disengagement reports. For the HDV group, we gathered data on Motor Vehicle Miles of Travel (MVMT) from the SWITRS Annual Report of Fatal and Injury Motor Vehicle Traffic Collisions for non-accident analysis. On the other hand, we also incorporated Annual Average Daily Traffic (AADT) data at specific accident sites from the California Traffic Census Program. The description can be seen in response to comment 4.

Comment 21:

line 358-359: Please replace "reasons are" by "reasons might be" since your analysis does not reveal the technical reasons for your observations and your explanations are just hypotheses.

Response:

We have revised line 358-359 to reflect your suggestion:

Line 442-443:

The possible reasons might be a lack of situational awareness in complex driving scenarios and limited driving experience of AVs ¹⁵.

REVIEWERS' COMMENTS

Reviewer #1 (Remarks to the Author):

Thank you very much to the authors, most of my previous comments were dealt with carefully, except my biggest concern. It also seems like reviewer 2 shares the same view as me. I understand that the authors are not comparing accidents vs non-accidents but the focus was on to exploring the differential characteristics of accidents involving AVs vs HDVs / comparing the distributions of AV accidents and HDV accidents over various situations.

Agreeing with review 2, this needs to be carefully interpreted. I can see the authors have tried their best on being careful. Given that the analysis can only conclude that, when accidents involving HDVs occur, it has x% chance that it happens under y condition. For example, 'accidents involving HDVs occur slightly more frequently under these conditions, at a rate of 83%, compared to 73% for AVs' or 'if we look at all HDV accidents, 83% took place under clear skies'. Perhaps the abstract can be even more explicit to avoid misunderstanding.

The following statement in abstract can be misunderstood, 'A matched case-control design was conducted to investigate the impact of different variables on the likelihood of accidents involving AV versus HDV.'. The phrase 'likelihood of accidents' sound like the tendency of accidents happening, which often takes into account of non-accidents. 'A matched case-control design was conducted to investigate the differential characteristics involving AVs vs HDVs.'

The following statement should in abstract should also be dealt with caution. 'The analysis suggests that AVs tend to be safer than HDVs in many accident situation.' I am not quite sure how this conclusion was drawn and what 'many accident situation' refers to here, does this refer to Severity of Accident Outcomes? It will be good to be explicit, because what I can see is that 2.2% of accidents involving AVs led to fatality, whereby 0.89% of accidents involving HDVs led to fatality.

Reviewer #2 (Remarks to the Author):

The second revision of the paper improved clearly over the first revision. My concerns have been addressed adequately. The paper is well organized and readable.

Concerning the explanations that you are giving in section 6 on the behavior of AVs I do not fully agree although your explanations are to some extent reasonable. For the present paper I believe your arguments are sufficient. However, just as an idea for future work, you could ask a group of experts on AVs what they believe which factors are responsible for the observed differences between HDVs and AVs and report their answers.

NCOMMS-23-46709

A Matched Case-Control Analysis of Autonomous vs Human-Driven Vehicle Accidents

Modifications and Responses to the Reviewers' Comments and Suggestions

The authors are grateful to the reviewers for their careful review and feedback. The manuscript was revised to address all the Reviewers' comments. For your convenience added or modified parts are marked **in blue** in the Response document.

Reviewer #1

General Comment:

Thank you very much to the authors, most of my previous comments were dealt with carefully, except my biggest concern. It also seems like reviewer 2 shares the same view as me. I understand that the authors are not comparing accidents vs non-accidents but the focus was on to exploring the differential characteristics of accidents involving AVs vs HDVs / comparing the distributions of AV accidents and HDV accidents over various situations.

Response:

We sincerely appreciate the time and effort you have dedicated to reviewing our manuscript and providing your valuable feedback. Your insights, along with those from Reviewer 2, have significantly contributed to enhancing the quality of our work. For your concern about accidents vs non-accidents, we implement the following measures to address your request: 1. A comparison of several factors including accident rates are calculated as shown in the supplement files to address your concern. 2. Direct comparison between AV and HDV accidents is still not viable as the difference in exposure and number of vehicles of both types are extremely unbalanced. 3. Non-accidents don't have any variables of interest to conduct a matched control design. We hope that these revisions and additions address your concerns and clarify the contributions of our study to the existing literature on AV vs HDV accidents.

Comment 1:

Agreeing with review 2, this needs to be carefully interpreted. I can see the authors have tried their best on being careful. Given that the analysis can only conclude that, when accidents involving HDVs occur, it has x% chance that it happens under y condition. For example, 'accidents involving HDVs occur slightly more frequently under these conditions, at a rate of 83%, compared to 73% for AVs' or 'if we look at all HDV accidents, 83% took place under clear skies'. Perhaps the abstract can be even more explicit to avoid misunderstanding.

Response:

In response to your suggestion, we revised the abstract and the whole manuscript to be even more explicit to avoid misunderstanding. Specifically in the abstract:

A matched case-control design was conducted to investigate the differential characteristics involving Autonomous Vehicles versus Human-Driven Vehicles accidents. The analysis suggests that Advanced Driving Systems generally have a lower chance of occurring than Human-Driven Vehicles in most of the similar accident scenarios. However, accidents involving Advanced Driving Systems occur more frequently than Human-Driven Vehicle accidents under dawn/dusk or turning conditions, which is 5.250 and 1.988 times higher, respectively.

Comment 2:

The following statement in abstract can be misunderstood, 'A matched case-control design was conducted to investigate the impact of different variables on the likelihood of accidents involving AV versus HDV.'. The phrase 'likelihood of accidents' sound like the tendency of accidents happening, which often takes into account of non-accidents. 'A matched case-control design was conducted to investigate the differential characteristics involving AVs vs HDVs.'

Response:

In response to your suggestion, we revised the abstract to be even more explicit to avoid misunderstanding. Specifically:

A matched case-control design was conducted to investigate the differential characteristics involving AVs vs HDVs accidents.

Comment 3:

The following statement should in abstract should also be dealt with caution. 'The analysis suggests that AVs tend to be safer than HDVs in many accident situation.' I am not quite sure how this conclusion was drawn and what 'many accident situation' refers to here, does this refer to Severity of Accident Outcomes? It will be good to be explicit, because what I can see is that 2.2% of accidents involving AVs led to fatality, whereby 0.89% of accidents involving HDVs led to fatality.

Response:

In response to your suggestion, we revised the abstract to be even more explicit to avoid misunderstanding. Specifically:

The analysis suggests that Advanced Driving Systems generally have a lower chance of occurring than Human-Driven Vehicles in most of the similar accident scenarios.

Reviewer #2

General Comment:

The second revision of the paper improved clearly over the first revision. My concerns have been addressed adequately. The paper is well organized and readable.

Response:

Thank you for your feedback. I'm glad to hear that the second revision of the paper has met your expectations and addressed the initial concerns effectively. Your insights are invaluable in the ongoing process of improving this work.

Comment 1:

Concerning the explanations that you are giving in section 6 on the behavior of AVs I do not fully agree although your explanations are to some extent reasonable. For the present paper I believe your arguments are sufficient. However, just as an idea for future work, you could ask a group of experts on AVs what they believe which factors are responsible for the observed differences between HDVs and AVs and report their answers.

Response:

The explanations in section 6 (Now in Discussion) are based on both accident distribution and the findings of the matched case-control study. This aims to mitigate potential controversy or overinterpretation.

For the future work idea, this research and its findings have garnered significant attention. The opinions of AV experts are collected from social media. The preprint of the paper has been viewed 144 times and downloaded 21 times. It was shared on LinkedIn by Dr. Philip Koopman from Carnegie Mellon University, where it received 187 likes, 24 comments, and 10 reposts (https://www.linkedin.com/posts/philip-koopman-0631a4116_pdf-a-matched-case-control-analysis-of-activity-7126885865653514241-ivt-/?utm_source=share&utm_medium=member_ios).

The full posts in LinkedIn are attached as the figure below:

[REDACTED]

Figure. Reposts and comments about this paper in LinkedIn

Additionally, the AVOID dataset used in the paper has been cited seven times.

Also, we added the following in the manuscript as future work:

Future research could also benefit from consulting a group of AV experts to identify and report on the factors contributing to the safety differences between HDVs and AVs. Reporting their responses could provide qualitative depth to the research findings.